# Reactive-site-centric chemoproteomics identifies a distinct class of deubiquitinase enzymes

David S. Hewings [1,2,3,4], Johanna Heideker[1,2], Taylur P. Ma[5], Andrew P. AhYoung[2], Farid El Oualid [6], Alessia Amore[6], Gregory T. Costakes[7], Daniel Kirchhofer [2], Bradley Brasher[7], Thomas Pillow [3], Nataliya Popovych[2], Till Maurer[8], Carsten Schwerdtfeger[7], William F. Forrest[9], Kebing Yu[5], John Flygare[3,10], Matthew Bogyo[4] & Ingrid E. Wertz[1,2]

Activity-based probes (ABPs) are widely used to monitor the activity of enzyme families in biological systems. Inferring enzyme activity from probe reactivity requires that the probe reacts with the enzyme at its active site; however, probe-labeling sites are rarely verified. Here we present an enhanced chemoproteomic approach to evaluate the activity and probe reactivity of deubiquitinase enzymes, using bioorthogonally tagged ABPs and a sequential on-bead digestion protocol to enhance the identification of probe-labeling sites. We confirm probe labeling of deubiquitinase catalytic Cys residues and reveal unexpected labeling of deubiquitinases on non-catalytic Cys residues and of non-deubiquitinase proteins. In doing so, we identify ZUFSP (ZUP1) as a previously unannotated deubiquitinase with high selectivity toward cleaving K63-linked chains. ZUFSP interacts with and modulates ubiquitination of the replication protein A (RPA) complex. Our reactive-site-centric chemoproteomics method is broadly applicable for identifying the reaction sites of covalent molecules, which may expand our understanding of enzymatic mechanisms.

[1] Discovery Oncology, Genentech, South San Francisco, California 94080, USA. [2] Early Discovery Biochemistry, Genentech, South San Francisco, California 94080, USA. [3] Discovery Chemistry, Genentech, South San Francisco, California 94080, USA. [4] Department of Pathology, Stanford University School of Medicine, Stanford, California 94305, USA. [5] Microchemistry, Proteomics and Lipidomics, Genentech, South San Francisco, CA 94080, USA. [6] UbiQ Bio BV, Science Park 408, 1098 XH Amsterdam, The Netherlands. [7] Boston Biochem Inc., 840 Memorial Drive, Cambridge, Massachussetts 02139, USA. [8] Structural Biology, Genentech, South San Francisco, California 94080, USA. [9] Bioinformatics, Genentech, South San Francisco, California 94080, USA. [10] Present address: Merck, 630 Gateway Boulevard, South San Francisco, California 94080, USA. These authors contributed equally: Taylur P. Ma and Andrew P. AhYoung. Correspondence and requests for materials should be addressed to I.E.W. (email: wertz.ingrid@gene.com)

Activity-based probes (ABPs) are powerful tools to study enzymatic activity in complex biological systems. These probes are typically small-molecule inhibitors or substrate mimics that react covalently with active-site residues, and permit detection of the active subpopulation of a class of enzymes[1,2]. ABPs are especially suited to the study of enzymes possessing nucleophilic active-site residues, as particular electrophilic reactive groups show selectivity toward certain catalytic nucleophiles (e.g., fluorophosphonates for serine hydrolases and vinyl sulfones (VSs) for cysteine proteases). Activity-based protein profiling (ABPP) uses ABPs to monitor the activity of many enzymes simultaneously, providing a snapshot of the active enzymes of a particular class. However, inferring enzyme activity from ABP reactivity relies on the assumptions that the probe and enzyme have undergone a covalent reaction, that the probe has reacted specifically at the active site, and that inactive enzymes will not react with the probe. Testing these assumptions is challenging and requires protein-by-protein validation, such as mutation of active site residues.

Deubiquitinases (DUBs) are proteases that cleave after the C terminus of ubiquitin (Ub) and have been widely studied by ABPs[3]. Ubiquitination regulates many essential intracellular processes in eukaryotes. Ub, an 8.5 kDa globular protein, can be attached to the N termini or Lys (K) side chains of target proteins via its C-terminal Gly residue. Ub may also be modified to form polyubiquitin chains. The canonical role of ubiquitination is to direct target proteins for proteasomal degradation, typically via K48-linked Ub chains[4]. Other linkage types have diverse functions, e.g., K63-linked chains may serve as protein assembly scaffolds[5]. Functionally, DUBs hydrolyze Ub chains, cleave Ub from substrates, or generate free Ub from its genetically encoded precursors[6]. The human genome encodes for 88 known active DUBs including 80 cysteine proteases (Supplementary Table 1). As all DUBs must bind Ub adjacent to their catalytic site, ABPs for DUBs typically consist of full-length Ub in which a Cys-reactive electrophilic group such as a VS, vinyl methyl ester (VME)[7,8] or propargylamide (PA)[9] replaces the C-terminal Gly residue. In addition, a reporter tag is appended at the N terminus. Since their introduction, DUB ABPs have been valuable for the identification of new DUB families[8,10], for characterizing DUB selectivity[11], for profiling changes in DUB activity across conditions or stimuli[12,13], and in drug discovery[14,15].

Previous reports indicate that DUB ABPs also label many non-DUB proteins. These proteins include enzymes that form a covalent thioester with the C terminus of Ub, and therefore their reaction with DUB ABPs is unsurprising[16]. However, other proteins are also enriched by probe treatment and affinity purification, even under denaturing conditions[8,16,17]. It is unclear whether these proteins react covalently with the probe or whether they are enriched due to non-covalent interaction with the probe, other probe-labeled proteins, or the purification matrix. Furthermore, the DUB OTUB1 reacts with Ub-VS on a non-catalytic Cys residue[18], suggesting that DUB ABPs may not always react with DUBs only at their active site. Taken together, some DUBs, which are presumed active on the basis of ABPP experiments, may in fact be identified due to reaction on non-catalytic sites, or through non-covalent interactions. Indeed, these limitations extend to any technique where a covalent reaction between a protein and a probe molecule is inferred without identifying the sites of reaction.

To overcome these limitations, we modified the design of Ub-based ABPs and optimized enrichment methodologies, in order to permit the detection of probe-labeled residues by mass spectrometry (MS). This approach confirms the expected labeling of DUB catalytic Cys residues by ABPs, as well as unexpectedly widespread labeling of non-catalytic Cys residues across DUB classes and of non-DUB proteins. Using this method, we identify zinc finger (ZnF) with UFM1-specific peptidase domain protein (ZUFSP), a previously unannotated DUB that represents a unique class of DUB enzymes.

## Results

**Detecting labeling sites of DUB ABPs by LC-MS/MS.** To characterize the sites of probe labeling in a DUB ABPP experiment using established DUB ABPs[8], we treated HEK 293T lysate with one of three hemagglutinin (HA)-tagged DUB ABPs, performed immunoprecipitations (IP) with anti-HA resin, and processed the enriched proteins using filter-assisted sample preparation (FASP)[19] followed by liquid chromatography and tandem MS (LC-MS/MS) analysis[14]. We detected probe modification only on Cys residues (Fig. 1a). Probe-modified peptides were detected for seven DUBs (Fig. 1b). Strikingly, we detected modification of UCHL1 and OTUB1 on both the active-site Cys residue and a non-catalytic Cys, whereas for USP14 we could detect only labeling on a non-catalytic site. Furthermore, modified peptides were detected for 11 non-DUB proteins (Supplementary Table 2). This suggests that labeling by DUB ABPs away from the active site may be rather widespread and, if so, these probes cannot truly be regarded as activity-based, as labeling on non-catalytic sites is not indicative of catalytic activity. However, as we could only detect probe-modified peptides for a small proportion of the total proteins identified in the HA-immunoprecipitated material, it was unclear whether labeling of DUBs on non-catalytic sites and covalent modification of non-DUB proteins by DUB ABPs is more prevalent.

**Developing a reactive-site-centric chemoproteomics method.** We developed a 'reactive-site-centric chemoproteomics' protocol, in order to identify a higher proportion of probe-labeling sites (Fig. 1c). First, we synthesized[20] and evaluated two bifunctional DUB ABPs (Fig. 2a and Supplementary Fig. 1–3): Ub vinyl pentynyl ester (VPE) and Ub vinyl pentynyl sulfone (Ub-VPS). Both contain an N-terminal HA tag for detection, residues 1–75 of human Ub as a recognition element, a Cys-reactive electrophile, and a C-terminal alkyne for attachment to a cleavable biotin-azide tag via 'click chemistry.' The ester moiety of the VPE probe was unstable during trypsin digestion (Supplementary Table 3) and thus only the VPS probe was used in further studies. Addition of the alkyne tag does not adversely affect probe reactivity, as Ub-VPS and Ub-VS show similar enrichment of DUBs after anti-HA IP (Fig. 2b).

Next, we required a biotin-azide tag that would be resistant to stringent washing and to digestion by multiple proteases, and could be efficiently cleaved to give labeled peptides[21]. The hydrazine-labile $N$-1-(4,4-dimethyl-2,6-dioxocyclohexylidene) ethyl (DDE) linker[22] (Fig. 2c) was unsuitable as no labeled peptides could be detected after hydrazine treatment, despite effective protein enrichment (Supplementary Fig. 4). This is likely to be due to instability of the linker during on-bead trypsin digestion (Supplementary Table 4). Instead, we turned to the acid-labile dialkoxydiphenylsilane (DADPS) linker[23–25]. This linker was suitable for on-bead digestion and identification of probe-labeling sites, as labeled peptides could be detected following enrichment and DADPS cleavage, as discussed below (Fig. 3).

To assess the effectiveness of biotin–streptavidin affinity purification and on-bead digestion, we compared the number of DUBs identified by this method with those identified by the established method of anti-HA IP[8] followed by FASP (Fig. 3a and Supplementary Data 1). For on-bead digestion, we used trypsin alone or Lys-C followed by trypsin: digestion with Lys-C under strongly denaturing conditions before trypsin digestion has been reported to increase digestion efficiency over digestion with

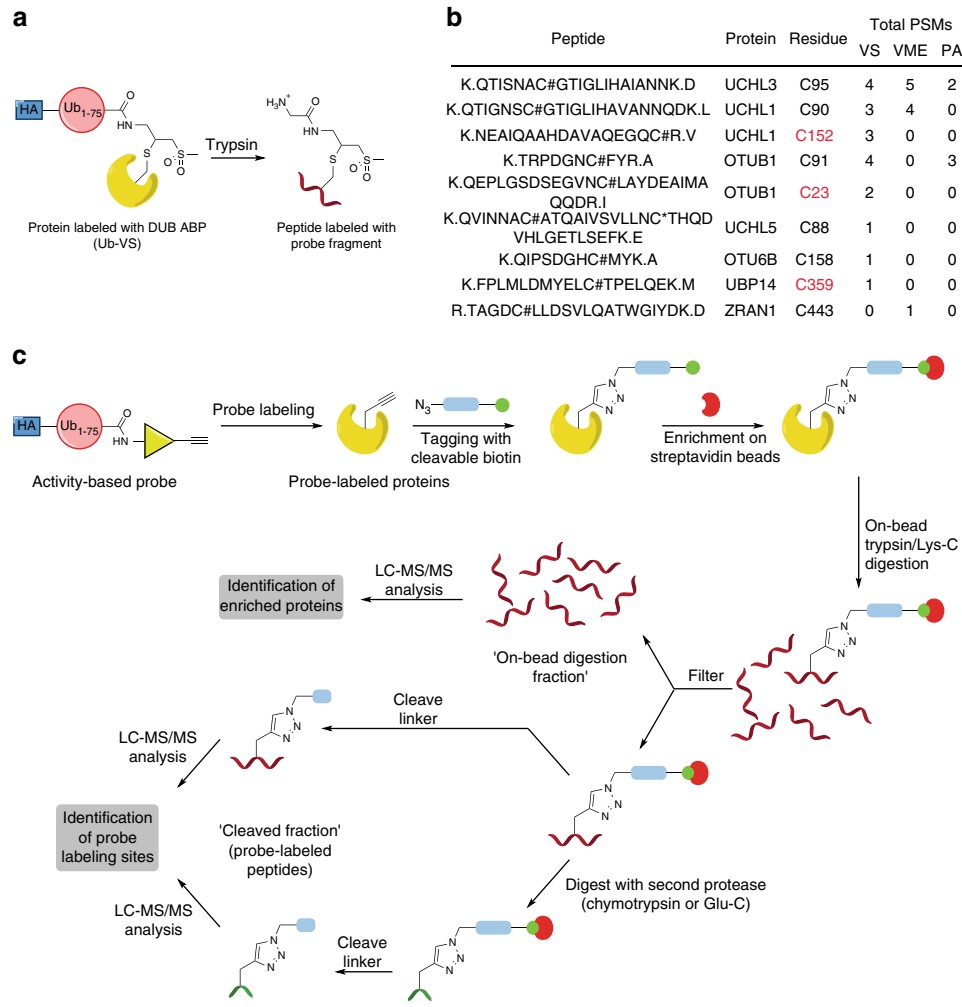

**Fig. 1** Conventional DUB ABPs label DUBs on both catalytic and non-catalytic Cys residues. **a** Structure of probe-labeled peptide fragment generated by trypsin digestion of Ub-VS-labeled proteins. In these ABPs, an HA tag is attached to residues 1–75 of ubiquitin by a spacer of two aminohexanoyl units. **b** Detection of probe-labeled DUB peptides in HEK 293T lysate after anti-HA immunoprecipitation and filter-assisted sample preparation (FASP), using HA-tagged Ub-VS, Ub-VME, or Ub-PA probes. Peptides were searched using the probe fragment as a variable modification on Cys. (Modification on other residues could not be detected.) Non-catalytic Cys residues are shown in red. # = Ub-probe fragment; * = carbamidomethyl modification; @ = methionine oxidation. **c** Reactive-site-centric chemoproteomic strategy to identify probe targets and labeling sites. Lysates are treated with an alkyne-containing ABP. Click chemistry is used to attach a cleavable biotin-azide tag and tagged proteins are enriched on streptavidin beads, using stringent washing to remove non-covalent binders. Proteins are digested on-bead to provide peptides for protein identification. The beads are then subject either to immediate linker cleavage to release probe-labeled peptides or to a second proteolysis before linker cleavage. The yellow triangle represents an electrophilic group, the pale blue rectangle represents a cleavable linker, and the green circle represents biotin

trypsin alone[26] and has been used for on-bead digestion without causing significant digestion of streptavidin[27]. Using equivalent inputs, all three enrichment/digestion methods identified similar numbers of DUBs from each class (Fig. 3a); thus, for identifying enriched DUBs, biotin–streptavidin affinity purification with on-bead digestion is similarly sensitive to anti-HA IP with FASP. Notably, all three methods identify JAB1/MPN/MOV34 (MPN +) DUBs, metalloproteases that are not expected to react with electrophilic warheads (Fig. 3a). Analysis of probe-labeled residues, combined with a suitable negative control to account for nonspecific enrichment, could clarify whether MPN + DUBs react covalently with DUB ABPs.

**Identifying selectively enriched proteins**. To account for non-specific enrichment due to background binding of proteins to streptavidin beads or promiscuity of the electrophile, we

developed a related ABP, SUMO2-VPS. This probe uses small ubiquitin-related modifier 2 (SUMO2) as a recognition element in place of Ub and thus should show minimal cross-reactivity with DUBs, but maintains the VS electrophile and alkyne handle for enrichment. This probe serves not only as a negative control (Supplementary Figs. 3 and 4) but also as an ABP for deSUMOy-lases (SENPs). To access SUMO2-VPS we developed a total linear chemical synthesis (described in the Supplementary Information) of the SUMO2 protein (Cys48 mutated to Ser, Supplementary Fig. 3)[28–31]. Gratifyingly, Ub-VPS strongly enriched for many Cys protease DUBs, whereas SUMO2-VPS enriched deSUMOylases (Fig. 3b and Supplementary Data 2). MPN + DUBs are not significantly enriched by either probe. SENP8, a deNED-Dylase, is modestly enriched by the Ub probe compared with the SUMO2 probe, consistent with the sequence similarity of NEDD8 and Ub. An alternative negative control, which employed the Ub-VPS probe but omitted CuSO4 during the tagging of labeled

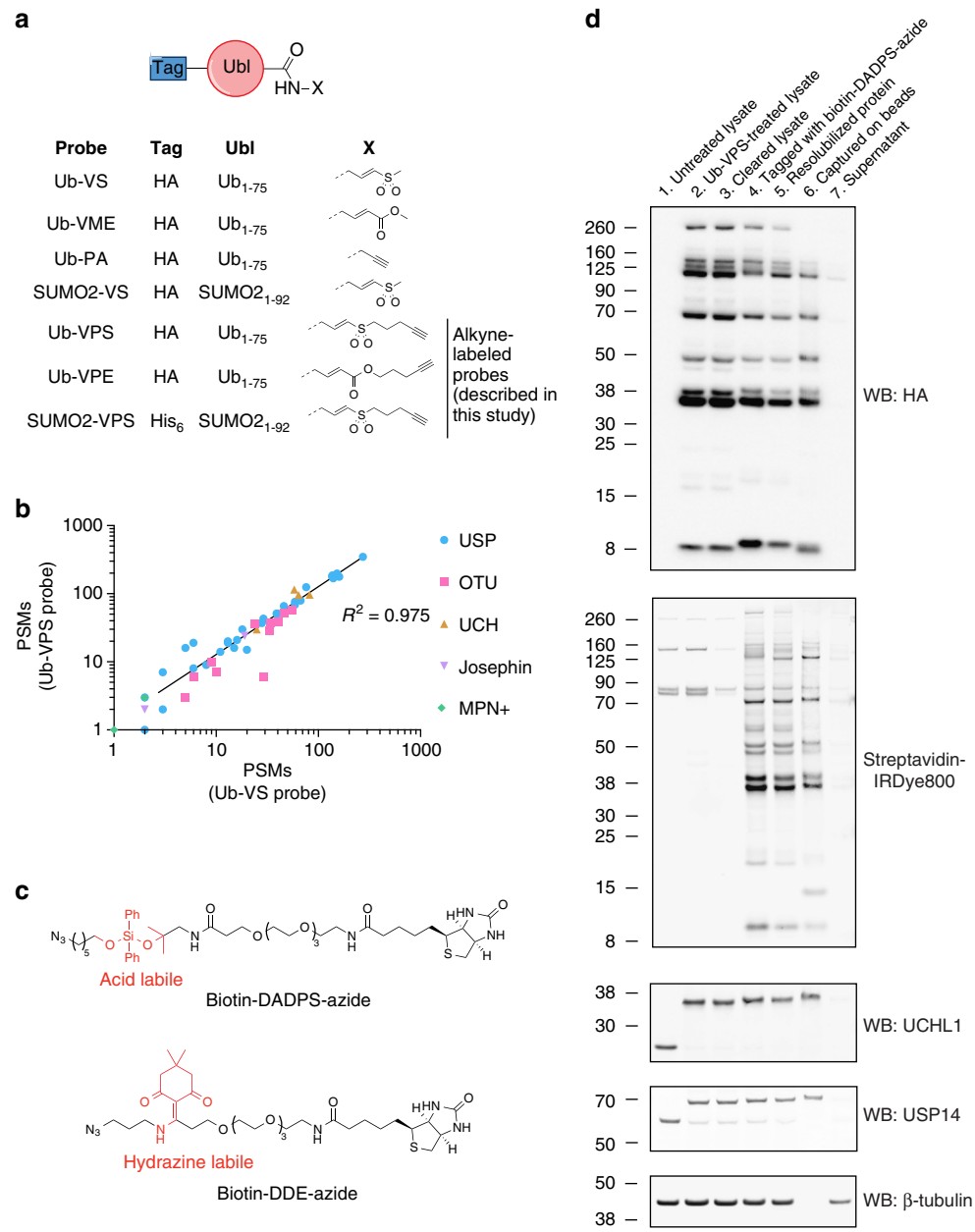

**Fig. 2** Design and reactivity of alkyne-containing DUB probes. **a** Structures of probes used in this work. The vinyl pentynyl sulfone (VPS) and vinyl pentynyl ester (VPE) probes maintain the electrophilic groups of the commonly-used vinyl sulfone (VS) and vinyl methyl ester (VME) probes, and additionally contain an alkyne for attachment to biotin. Ubl, ubiquitin/ubiquitin-like protein. **b** Peptide-spectrum matches (PSMs) for DUBs detected following anti-HA IP, using either Ub-VS or Ub-VPS probes. Data are representative of two biological replicates. **c** Structures of linkers used in this work. **d** Western blottings (or streptavidin-biotin blotting) indicate the effectiveness of labeling with Ub-VPS, tagging with biotin-DADPS-azide, precipitation, and enrichment

proteins with biotin-DADPS-azide, also enriched DUBs over known non-DUB proteins (Supplementary Fig. 5 and Supplementary Data 2).

To evaluate the sensitivity of our method, we compared DUB detection, by ABPP with Ub-VPS, to DUB expression, as indicated by RNA sequencing (RNA-Seq) (Supplementary Fig. 6 and Supplementary Data 3). Of the 73 Cys protease DUBs expressed in HEK 293T cells (reads per kilobase per million mapped reads (RPKM) > 1), 60 were detected by ABPP. Of the 13 not detected, several are known to be inactive toward Ub substrates (USP18, USP53, PAN2, and ALG13) or have been reported to show little reactivity toward mono-Ub ABPs (TNFAIP3 and OTULIN). Three out of four MINDY DUBs

were not detected by ABPP, despite being expressed, likely to be due to their strong preference for long Ub chains[32]. OTUB1, USP13, and the inactive USP39 (SNUT2) were detected with Ub-VPS but showed only modest enrichment over the SUMO2-VPS control, perhaps indicative of nonspecific interaction with the resin. USP54, which is annotated as an inactive DUB, is detected at low levels with the Ub-VPS probe, consistent with a catalytic domain that possesses the catalytic triad but lacks other conserved residues important for interaction with the C terminus of Ub[33]. Thus, our method can detect the majority (~ 90%) of expressed, active DUBs in HEK 293T cells, selectively enriches for active DUBs, and can distinguish bona fide enrichment by the probe from nonspecific interactions.

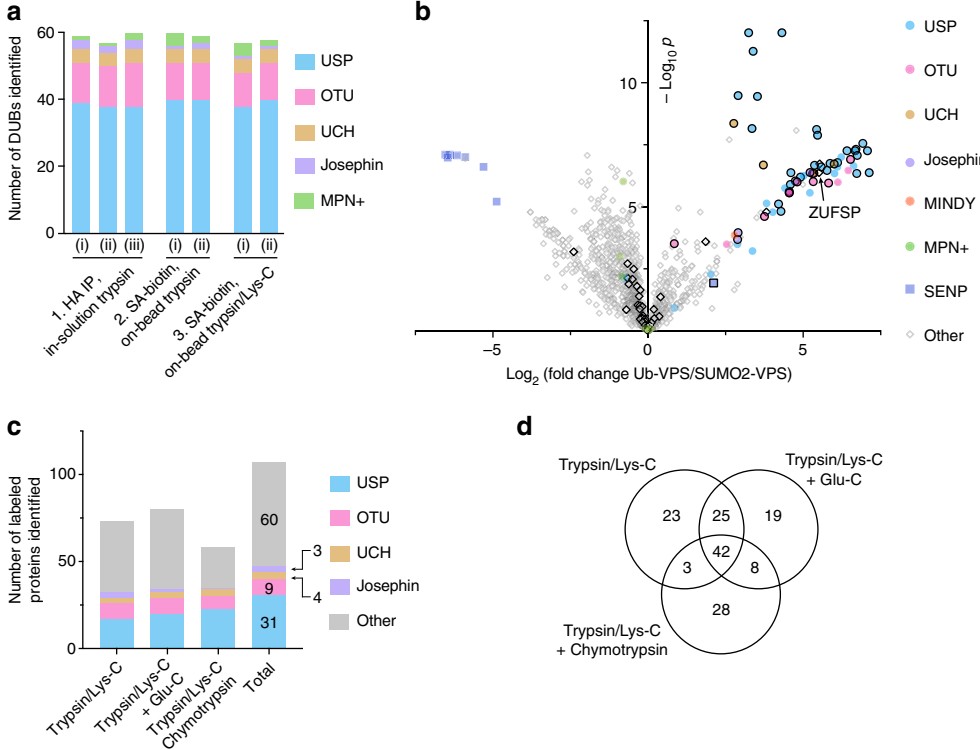

**Fig. 3** A reactive-site-centric chemoproteomic method to identify probe-labeling sites. **a** Comparison of the number of DUBs identified by different purification or digestion methods. HEK 293T lysate was treated with Ub-VS (condition 1) or Ub-VPS (conditions 2 and 3), and then subject to anti-HA IP followed by FASP using trypsin (condition 1) or tagged with biotin-DADPS-azide, enriched on streptavidin (SA) beads, and digested on-bead with trypsin (condition 2) or Lys-C and trypsin (condition 3). Shown is the number of DUBs identified with ≥ 3 unique peptides, in each of two or three biological replicates (indicated with Roman numerals). **b** Volcano plot of pairwise comparison of proteins in on-bead digestion fraction from ABPP using Ub-VPS probe and SUMO2-VPS probe. Statistical significance ($-\log_{10}$ p-value) is plotted against fold enrichment (average $\log_2$) from three biological replicates. A black border indicates proteins with high-confidence probe-labeling sites (i.e., sites identified in more than one experiment). ZUFSP is indicated with an arrow. **c** Comparison of the number of labeled proteins identified after cleavage of DADPS linker using different combinations of proteases. Shown is the number of proteins for which one or more labeled peptides were detected in any one of six biological replicates. Each replicate corresponds to a nominal protein input of 3 mg. **d** Venn diagram showing the intersections of probe-labeling sites identified using different combinations of proteases. Diagram produced using Venny 2.1 (http://bioinfogp.cnb.csic.es/tools/venny/)

**Mapping probe-labeling sites**. To identify Ub-VPS-labeled sites, we analyzed the peptides released after on-bead digestion and DADPS cleavage. Previous studies using enzymatic digestion followed by linker cleavage to identify modified peptides[24,34–38] employed only trypsin to generate peptides for MS identification. As labeling may occur at sites that do not give tryptic peptides of suitable length for LC-MS/MS detection, we explored the use of sequential on-bead digestion with multiple proteolytic enzymes to improve the coverage of labeling sites (Fig. 1c). Preliminary experiments suggested that an initial digestion with trypsin/Lys-C, followed by removal of the 'on-bead digestion fraction,' and a second digestion with Glu-C or chymotrypsin before linker cleavage identified a different subset of labeling sites from trypsin or trypsin/Lys-C alone (Supplementary Table 5 and Supplementary Fig. 7).

By combining our optimized enrichment strategy with sequential on-bead digestion, we identified high-confidence probe-labeled sites for 43 DUBs (Fig. 3b and Table 1). Different protease combinations identify distinct subsets of probe-labeling sites (Fig. 3c, d), confirming the benefit of sequential on-bead digestion. Compared with the in silico-generated peptide sequences assuming full tryptic cleavage, peptides identified after combined cleavage with trypsin/Lys-C + chymotrypsin or trypsin/Lys-C + Glu-C are shorter, and many would be missed by trypsin/Lys-C treatment alone (Supplementary Fig. 8). The benefit of sequential on-bead digestion

was particularly apparent for USP DUBs, which often have long tryptic peptides spanning their active site: trypsin/Lys-C alone identified labeling sites for only 17 USP DUBs, but additional digestions with Glu-C or chymotrypsin increased detection to 31 (27 with high-confidence sites). The MPN + DUBs, which are detected using traditional purification methods (Fig. 3a), were not significantly enriched and no labeled peptides were detected (Fig. 3b). A full list of labeling sites identified under each digestion condition is presented in Supplementary Data 4. We were unable to identify labeling sites of SUMO2-VPS probe under any of the digestion conditions, likely due to the large size and complex fragmentation patterns produced by the SUMO2 C-terminal fragment that remains attached to the labeled peptide after proteolytic digestion.

**Ub-VPS labels some DUBs on multiple sites**. Of the 43 DUBs with high-confidence labeling sites, we identified non-catalytic labeling sites for 11 DUBs (Table 1). For UCHL1, four labeling sites were identified (Fig. 4a and Supplementary Data 4), including the catalytic site (C90). Annotated MS/MS spectra supporting the assignment of these labeling sites are provided in Supplementary Fig. 9. Mapping the labeled Cys residues to an X-ray crystal structure of UCHL1 bound to Ub-VME[39] (Fig. 4b)

**Table 1 | Ub-VPS-labeling sites identified in HEK 293T cell lysate**

| Class | Protein | Labeling sites | Class | Protein | Labeling sites |
|---|---|---|---|---|---|
| USP | UBP1 | **C90** | Other | 1433T | C134 |
| | UBP3 | **C168** | | A6NLJ7 | C9 |
| | UBP5 | C219*, **C335**, **C471*** | | ADRM1 | C357 |
| | UBP7 | C90*, C121*, **C223**, **C315*** | | ALDOA | **C339** |
| | UBP8 | **C786**, C809* | | AUP1 | C391 |
| | USP9X | **C1566** | | BOLA2 | C31 |
| | UBP10 | **C424**, **C436***, C697* | | CATB | C319 |
| | UBP12 | **C48** | | CH60 | **C442** |
| | UBP13 | C345 | | CLIC4 | C35 |
| | UBP14 | C105*, **C114**, C122*, **C203***, **C257***, **C359***, C415* | | DCAF7 | **C256** |
| | UBP15 | C289*, **C298** | | DDX55 | **C437** |
| | UBP16 | **C205** | | DFFA | C165 |
| | UBP19 | **C506** | | EF1A1 | **C411** |
| | UBP21 | C221 | | EF2 | C41 |
| | UBP22 | **C171***, **C185** | | EIF3I | **C76** |
| | UBP24 | **C1698** | | FETUA | **C132** |
| | UBP25 | **C178** | | HECD1 | **C2579** |
| | UBP27 | **C87**, **C91*** | | IF5A1 | **C22**, C73 |
| | UBP28 | **C171** | | IMDH2 | **C331** |
| | UBP30 | **C77** | | KLH28 | C189 |
| | UBP33 | **C194** | | KPYM | C423, **C424** |
| | UBP34 | **C1903** | | LDH6A | **C163** |
| | UBP35 | **C450** | | LDHA | C35 |
| | UBP36 | C131 | | LDHB | C36, **C294** |
| | UBP37 | **C350** | | MIF | C60, **C81** |
| | UBP38 | **C454** | | NDK8 | C94 |
| | UBP40 | **C50** | | PAIRB | C11 |
| | UBP42 | **C120** | | PAL4A | **C62** |
| | UBP45 | **C199** | | PCBP1 | **C54** |
| | UBP47 | **C197** | | PPIA | **C115** |
| | UBP54 | C42 | | PROF1 | **C128** |
| OTU | OTU1 | **C160** | | Q5HYB6 | **C154** |
| | OTU6B | **C158**, **C172*** | | RCC2 | **C428** |
| | OTU7A | **C210** | | RHOA | **C16** |
| | OTUB1 | **C91** | | RL10L | C105 |
| | OTUB2 | **C51** | | RL12 | **C141** |
| | OTUD3 | **C76** | | RL14 | C42 |
| | OTUD4 | **C45** | | RL23 | **C28** |
| | OTUD5 | **C224** | | RL9 | **C134** |
| | ZRAN1 | **C443** | | RS20 | **C70** |
| UCH | BAP1 | **C91**, C103* | | RS27L | **C77** |
| | UCHL1 | **C47***, **C90**, **C152***, **C220*** | | RS28 | **C27** |
| | UCHL3 | **C50***, **C95** | | RS3 | C134 |
| | UCHL5 | **C9***, **C88**, **C100*** | | RS6 | C12 |
| JOS | ATX3 | **C14**, **C18*** | | RSSA | C148 |
| | JOS1 | **C36** | | SAP | C409, C476 |
| | JOS2 | **C24** | | SENP8 | **C163**† |
| | | | | SERA | **C369** |
| | | | | SPEE | **C71** |
| | | | | SQSTM | C113 |
| | | | | TBA1A | **C25**, **C129**, **C295**, C347, **C376** |
| | | | | TBA1B | **C347** |
| | | | | TBB2A | **C12**, **C239**, **C303**, **C354** |
| | | | | TBB4A | C127 |
| | | | | TBC15 | **C24** |
| | | | | TEBP | **C40** |
| | | | | THIO | **C32**, C35, **C73** |
| | | | | UBE3C | **C1051**† |
| | | | | ZNFX1 | **C1860** |
| | | | | ZUFSP | **C360** |

High-confidence sites (bold) were detected in more than one experiment (out of six biological replicates, each employing three digestion conditions). *Non-active-site Cys residue of DUB; †catalytic residue of non-DUB. See Supplementary Data 4 for details of individual replicates.

revealed that one additional labeling site (C152) is in close proximity to the catalytic site, suggesting that the probe can react with either C90 or C152 when bound to the typical Ub-binding site[40]. The other two sites (C47 and C220) are far from the catalytic site and buried. To validate labeling of UCHL1 on C90 and C152, we purified the two single mutants UCHL1^C90A and UCHL1^C152A, and the double mutant UCHL1^C90A,C152A (Supplementary Fig. 10), and monitored their labeling with Ub-VS

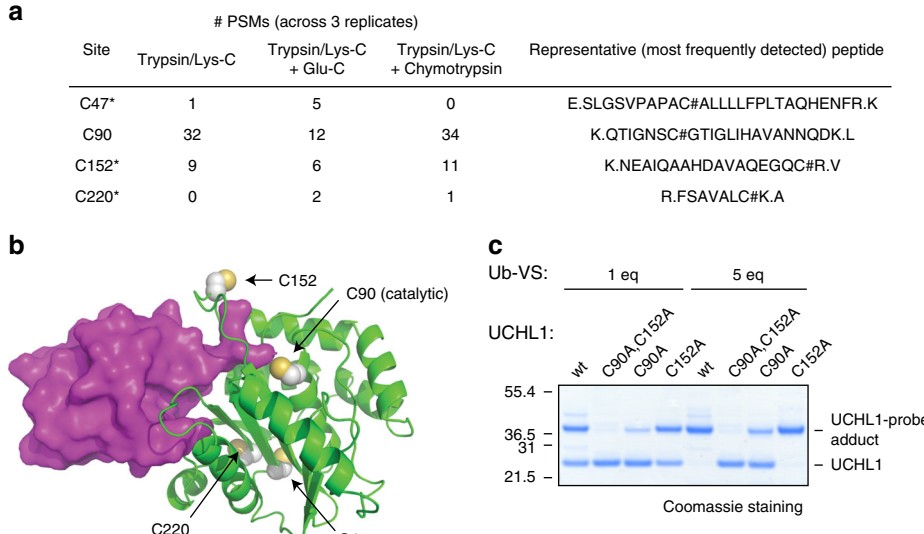

| Site | # PSMs (across 3 replicates) | | | Representative (most frequently detected) peptide |
|------|----------------|----------------------------|------------------------------|---------------------------------------------------|
| | Trypsin/Lys-C | Trypsin/Lys-C + Glu-C | Trypsin/Lys-C + Chymotrypsin | |
| C47* | 1 | 5 | 0 | E.SLGSVPAPAC#ALLLLFPLTAQHENFR.K |
| C90 | 32 | 12 | 34 | K.QTIGNSC#GTIGLIHAVANNQDK.L |
| C152* | 9 | 6 | 11 | K.NEAIQAAHDAVAQEGQC#R.V |
| C220* | 0 | 2 | 1 | R.FSAVALC#K.A |

**Fig. 4** Ub-VPS labels UCHL1 on multiple sites. **a** Sites of UCHL1 labeling identified by ABPP with Ub-VPS using different digestion conditions. **b** X-ray crystal structure of UCHL1 (green) bound to Ub-VME (purple). PDB ID: 3ifw. Labeled Cys residues are indicated in space-filling representation. **c** Reactivity of recombinant UCHL1 mutants toward Ub-VS. UCHL1 or the indicated mutants were treated with probe, then analyzed by SDS-PAGE and stained with Coomassie Brilliant Blue

(Fig. 4c). Both UCHL1[C90A] and UCHL1[C152A] were substantially labeled, whereas labeling was largely abolished in UCHL1[C90A, C152A]. The most significant probe reactivity therefore occurs on the catalytic Cys and an adjacent Cys residue. In a Ub 7-amido-4-methylcoumarin (Ub-AMC) hydrolysis assay[41], UCHL1[C152A] showed similar activity to the wild type, as previously reported[42], whereas UCHL1[C90A] and UCHL1[C90A,C152A] were inactive (Supplementary Fig. 11). Probe labeling of catalytically inactive UCHL1[C90A] substantiates the idea that probe reactivity is not necessarily indicative of DUB catalytic activity and highlights the importance of characterizing probe-labeling sites.

**Identification of ZUFSP as a DUB**. We also detected high-confidence probe-labeling sites on 40 proteins that are not annotated as DUBs. These included peptides from a number of highly abundant proteins including tubulins (which are labeled on several Cys residues of known high reactivity[43]), metabolic enzymes (such as lactate dehydrogenase), and ribosomal proteins. HECTD1 and UBE3C, which are HECT or RBR-type Ub ligases, respectively, are also labeled on their catalytic Cys residues, consistent with the proximity of a nucleophilic Cys residue to a Ub-binding domain. We were particularly intrigued by the labeling of ZUFSP, a protein of unknown function that was also significantly enriched at the protein level in the on-bead digestion fraction (Fig. 3b). Domain predictions[44] suggest that ZUFSP contains a Cys protease domain (residues 336–551) and four N-terminal $C_2H_2$-type ZnFs (Fig. 5a). Given that many DUBs are Cys proteases, and that certain ZnFs act as Ub-binding domains[45], we investigated whether ZUFSP may have deubiquitinating activity.

A BLAST search of the catalytic region indicated no homology to known DUB classes, but 40% similarity with UfSP2, a protease that cleaves at the C terminus of UFM1, a Ub-like protein[46] (Supplementary Fig. 12). Probe labeling was detected on C360 of ZUFSP (Fig. 5b), which aligns with the catalytic Cys of UfSP2. However, the alignment suggests that ZUFSP lacks the His residue required to form an active catalytic triad, and ZUFSP is predicted to be enzymatically inactive in the UNIPROT database. We built a homology model of the ZUFSP catalytic domain based

on the published X-ray crystal structure of mouse UfSP2[47] (Supplementary Fig. 13). The model indicated that H491 is positioned within 5 Å of the catalytic Cys residue and may therefore complete the catalytic triad.

To characterize ZUFSP binding to Ub and Ub-like proteins, we first expressed FLAG-tagged full-length ZUFSP and its C-terminal putative protease domain (residues 298–578), which lack the ZnFs, in HEK 293T cells. Pull-down experiments in lysate using Ub-, SUMO2-, or UFM1-conjugated agarose revealed that FLAG-ZUFSP and FLAG-ZUFSP(298–578) interacted with Ub-agarose, but not with UFM1- or SUMO2-agarose, suggesting that ZUFSP binds preferentially to Ub (Fig. 5c). Full-length ZUFSP was pulled down more effectively, suggesting that the N-terminal region may contribute to Ub binding.

To investigate the probe reactivity of ZUFSP, we purified full-length FLAG-tagged ZUFSP and two point mutants (C360A and H491A) from HEK 293T cells (Supplementary Fig. 14). We also purified recombinant full-length His₆-tagged ZUFSP and the corresponding C360A mutant from *Escherichia coli* (Supplementary Fig. 14). Wild-type ZUFSP reacted with Ub ABPs but not SUMO2-VS (S2VS), indicating that the reaction is dependent on Ub binding and is not simply due to the VS electrophile (Fig. 5d and Supplementary Fig. 15). ZUFSP[C360A] showed very little activity toward ABPs, corroborating the LC-MS/MS data that C360 is the primary Ub-probe-labeling site. Notably, ZUFSP[H491A] was also labeled by Ub probes, indicating H491 is not critical for probe reactivity (Fig. 5d). We also expressed the putative protease domain residues 298–578 and the corresponding C360A mutant in *E. coli* (Supplementary Fig. 14). Wild-type ZUFSP(298–578) but not the C360A mutant reacted with Ub ABPs (Fig. 5e), demonstrating that the N terminus is not required for probe reactivity. Thus, Ub ABPs label ZUFSP on C360, and neither the H491 residue nor the N-terminal region is required for probe reactivity.

To investigate whether ZUFSP has deubiquitinating activity, we reacted ZUFSP with di- or tetra-Ub chains containing various linkages. Full-length ZUFSP, whether expressed in HEK 293T cells or in *E. coli*, cleaved tetra-Ub K63-linked chains but showed little activity toward other chain types (Fig. 6a,b). Recombinant ZUFSP[C360A]-His₆ did not cleave any chain type,

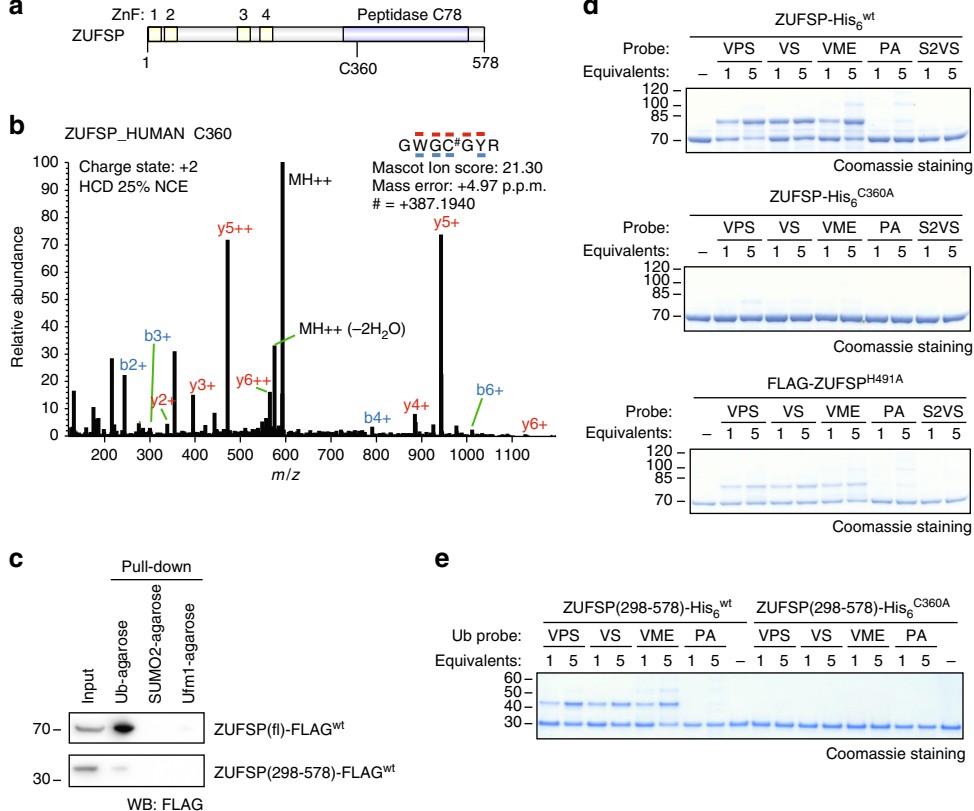

**Fig. 5** ZUFSP interacts with ubiquitin and reacts with Ub ABPs. **a** Domain structure of ZUFSP. Domain prediction from SMART (http://smart.embl-heidelberg.de)[44]. Figure produced using Illustrator for Biological Sequences (http://ibs.biocuckoo.org)[70]. **b** A representative MS/MS spectrum showing a peptide-spectrum matched to ZUFSP Cys-360-labeled peptide sequence. **c** Pull-down of FLAG-ZUFSP by Ub, SUMO2, or UFM1-conjugated agarose. Lysates from FLAG-ZUFSP-overexpressing cells were enriched on the indicated resins and analyzed by anti-FLAG immunoblotting. **d** Reactivity of full-length ZUFSP toward ABPs. Purified ZUFSP (wt, C360A, or H491A mutant) was treated with the indicated ABP (Ub-VPS, Ub-VS, Ub-VME, Ub-PA, or SUMO2-VS [S2VS]), then analyzed by SDS-PAGE and stained with Coomassie Brilliant Blue. **e** Reactivity of the catalytic domain of ZUFSP toward ABPs. Purified ZUFSP(298–578) (wt or C360A mutant) was treated with ABP, then analyzed by SDS-PAGE, and stained with Coomassie Brilliant Blue

indicating that C360 is the catalytic Cys residue. FLAG-ZUFSP[C360A] from HEK 293T cells showed very little activity; however, as the recombinant protein was completely inactive, we attribute the residual activity to trace amounts of co-purified DUBs. FLAG-ZUFSP[H491A] also showed very little activity toward Ub chains (comparable to FLAG-ZUFSP[C360A]), in contrast to the probe reactivity assays. This suggests that H491 is indeed an important residue for catalysis and, similar to UCHL1, provides another example to illustrate that probe labeling does not necessarily imply DUB activity (Fig. 6b). The recombinant ZUFSP catalytic domain showed only weak activity toward K63-linked tetra-Ub chains (Fig. 6c) and no activity toward any di-Ub chains (Supplementary Fig. 16), suggesting that the N-terminal region, which includes the ZnFs, facilitates Ub chain depolymerization. Full-length ZUFSP showed weak activity toward K63-linked di-Ub, and no activity towards other di-Ub linkage types (Supplementary Fig. 16), suggesting that ZUFSP has a preference for longer chains. Thus, we conclude that ZUFSP is a K63-selective DUB with a preference for longer chains, that C360 is the catalytic cysteine residue, and that both the H491 residue and the N-terminal regions are important for full DUB activity.

To further explore the catalytic activity of ZUFSP toward Ub chains, we combined ZUFSP with either a K63-linked or K48-linked tetra-Ub-modified 5-carboxytetramethylrhodamine (TAMRA)-conjugated peptide[14], and monitored the formation of the resulting TAMRA-conjugated products by in-gel fluorescence (Fig. 6d). ZUFSP rapidly processed the K63-linked Ub$_4$-peptide-TAMRA conjugate to Ub$_2$-peptide-TAMRA and Ub-peptide-TAMRA. Interestingly, only low levels of Ub$_3$-peptide-TAMRA were detected, suggesting that ZUFSP requires at least two Ub residues distal to the scissile bond to efficiently process the substrate. ZUFSP can clearly process Ub$_4$-containing species to Ub$_3$ (Fig. 6a), but Ub$_3$ formed by cleavage between the two proximal Ub moieties of Ub$_4$-peptide-TAMRA is not detected by this method, as it lacks a fluorophore. ZUFSP showed no hydrolytic activity toward the K48-linked substrate, suggesting that the enzyme recognizes the linkage type of the distal chain. The catalytic domain alone showed no activity toward either substrate (Supplementary Fig. 17), consistent with the minimal activity observed with unmodified Ub chains (Fig. 6c).

To facilitate further biochemical studies we sought suitable fluorogenic substrates for ZUFSP. The mono-Ub-containing substrate Ub-AMC[41] was not processed by ZUFSP, even at equal enzyme and substrate concentrations (250 nM) (Supplementary Fig. 11), confirming that ZUFSP is unable to process short chains. By contrast, ZUFSP processed K63-linked Ub$_4$-rhodamine 110 (Ub$_4$-Rh110), although it showed lower activity toward this substrate than USP5 (Supplementary Fig. 18a), which is highly active toward K63-linked chains[48]. We also monitored the cleavage of Ub$_4$-Rh110 by SDS-polyacrylamide gel electrophoresis (PAGE) and silver staining (Supplementary Fig. 18b). Two closely spaced bands appeared at the expected molecular weight of Ub$_2$, which we assign to free Ub$_2$ and Ub$_2$-Rh110, confirming that ZUFSP can cleave in the middle of the chain (endo cleavage).

Some DUBs show differences between their preferences for binding Ub modifications and their enzymatic selectivity[49]. It was therefore of interest to investigate whether ZUFSP could interact with other chain types. Surprisingly, nuclear magnetic resonance (NMR) studies revealed that full-length ZUFSP interacted with both Ub moieties of labeled K63-linked and K48-linked di-Ub (Fig. 6e and Supplementary Data 5). The catalytic domain alone showed no appreciable interaction with K48- or K63-linked di-

Ub, or with mono-Ub. Furthermore, K48-linked chains and, to a lesser extent, linear or mono-Ub inhibited processing of K63-linked Ub4-Rh110 by ZUFSP (Supplementary Fig. 19). Next, we expressed individual [15]N-labeled ZnFs to further investigate the contribution of the ZnFs to poly-Ub binding. NMR studies revealed that, in isolation, ZnFs 2, 3, and 4 all interact with mono-Ub, and with linear, K48- and K63-linked di-Ub (Supplementary Fig. 20). ZnF1, however, showed no appreciable interaction with

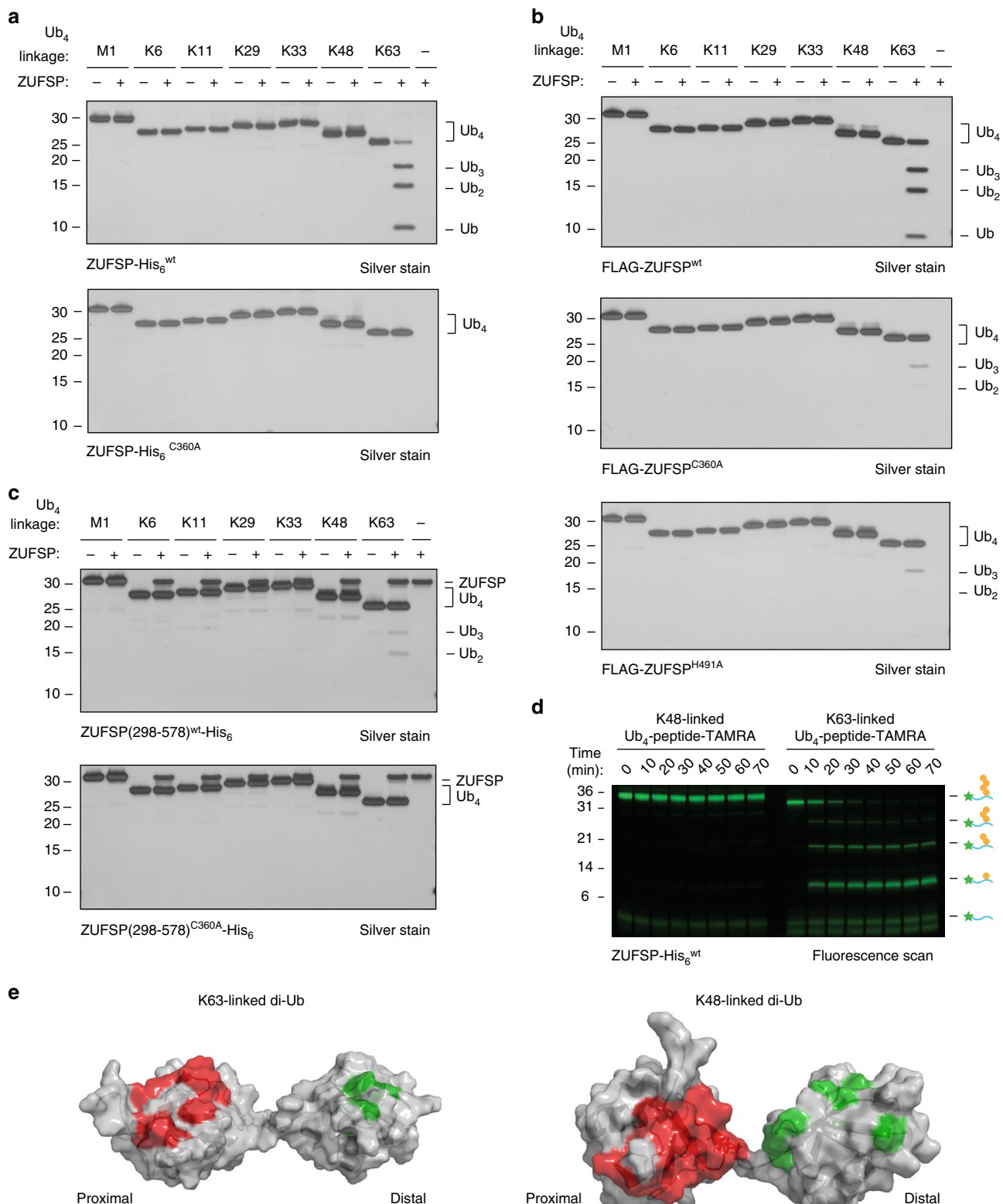

mono-Ub or any di-Ub chains tested. Given these findings, and because full-length ZUFSP is also able to bind several types of Ub chains, it is plausible that the catalytic domain imparts K63 linkage selectivity. Indeed, the ZUFSP catalytic domain alone, while showing very low enzymatic activity, appeared selective for K63 linkages (Fig. 6c).

Finally, we explored the cellular role of ZUFSP. ZUFSP is reported as a potential binding partner for the replication protein A (RPA) complex[50], which is involved in DNA replication and repair[51]. Analysis of affinity-purified FLAG-ZUFSP by LC-MS/MS (Supplementary Data 6) and western blotting (Fig. 7a) confirmed that RPA proteins co-purify with ZUFSP. RPA proteins are known to be ubiquitinated with K63 chains in response to replication stress[52,53]. To investigate whether ZUFSP regulates RPA ubiquitination, we knocked down ZUFSP in HEK 293T cells, and used linkage-specific antibodies[54,55] to immuno-precipitate K63-linked polyubiquitin after induction of replication stress with hydroxyurea (HU) (Fig. 7b). Both HU treatment and ZUFSP RNA interference (RNAi) increased K63 ubiquitination of endogenous RPA1 and RPA2. More specifically, ZUFSP RNAi enhanced RPA1 ubiquitination after HU treatment, whereas RPA2 ubiquitination was enhanced under basal conditions. ZUFSP RNAi also enhanced K63 ubiquitination of RPA1/2 after HU treatment in HeLa cells (Supplementary Fig. 21). K63 poyubiquitination of PARP1 and ATR, which both associate with RPA proteins, was less significantly affected by ZUFSP RNAi (Supplementary Fig. 21). Thus, endogenous ZUFSP interacts with and modulates the ubiquitination status of RPA complex proteins, further implicating ZUFSP as a bona fide cellular DUB.

## Discussion

It has been previously noted that chemoproteomics studies using covalent chemical probes rarely identify the sites of modification by the probes, due to difficulties in enriching or detecting the modified peptides. Yet, when these data are available, they greatly increase confidence in the hits identified and provide valuable information about the probe-binding modes and reaction sites[56]. Here we use 'reactive-site-centric chemoproteomics' to enrich and identify probe-reactive proteins. The two key elements of this method include: (1) an alkyne-tagged probe and an acid-cleavable biotin-DADPS-azide linker, which allow enrichment of labeled proteins, and (2) on-bead digestion and subsequent linker clea-vage to release labeled peptides for identification by LC-MS/MS. Importantly, we employ sequential on-bead digestion to 'trim' long tryptic peptides and make them more suitable for identifi-cation by LC-MS/MS. On-bead digestion with proteases other than trypsin has been suggested in order to generate labeled peptides of a more suitable length for analysis[34], and alternative proteases have been used on purified proteins to map covalent modification sites[57]. Our results indicate that multiple successive protease treatments enhance the detection of covalent modifica-tion sites. We envisage that this reactive-site-centric chemopro-teomics method, using probes tagged with bioorthogonal chemical reporters[58] followed by sequential on-bead digestion, will be broadly applicable for the identification of modification sites using other covalent probes.

By combining an alkyne-tagged DUB probe with our enrichment and digestion strategy, we demonstrated that all DUBs for which labeled peptides are detected do react at their active site residues with an ABP, Ub-VPS. However, we also found that non-catalytic residues of 11 DUBs react covalently with the probe. Although all DUBs labeled on non-catalytic residues were also labeled on cata-lytic residues we nevertheless recommend caution in inferring activity from probe labeling. Most non-catalytic labeling sites are within catalytic domains, suggesting that the probe binds in the catalytic domain but reacts with a non-catalytic residue. For example, UCHL1 reacts substantially at C152, a non-catalytic residue positioned close to the active site that has previously been shown to be highly reactive toward lipid-derived electrophiles[42,59]. A smaller number of labeling sites are found outside catalytic domains, which may arise from interactions between the probe and another Ub-binding site, as has been previously observed for OTUB1[18]. Indeed, the labeled non-catalytic residue C219 of USP5 is within a Ub-binding UBP-type ZnF and a crystal structure of the USP5 UBP with Ub suggests that the C terminus of Ub and C219 of USP5 are in close proximity[60].

Because of the nature of the MS experiment, we are unlikely to have identified all labeling sites for the probe-reacted proteins, due to low protein abundance or technical limitations such as poorly ionizing peptides. Indeed, we were unable to identify labeling sites for several strongly enriched DUBs such as USP36 and USP48. Nonetheless, site-level data on probe-labeled residues and protein-level enrichment data (relative to a suitable control) provide com-plementary information to increase confidence in the hits obtained from ABPP and reduce false positives. For example, MPN + DUBs, which are not expected to react in an activity-dependent manner with the ABP due to the lack of a catalytic Cys, are consistently identified by IP-based DUB ABPP methods (Fig. 3a and Supple-mentary Data 1). We are unable to detect probe-labeling sites on these proteins and they are not enriched relative to a negative control probe (SUMO2-VPS) or a control in which labeled proteins are not tagged with biotin. This strongly suggests that MPN + DUBs do not react covalently with Ub-VPS, nor are they enriched due to non-covalent interaction with the probe, and instead are detected due to nonspecific interactions with the resin.

We identified high-confidence probe-labeling sites on similar numbers of non-DUB proteins (40) and DUBs (43). The list of labeled non-DUBs (Table 1) is dominated by highly abundant proteins, such as tubulin and ribosomal proteins, which may not bind Ub but possess sufficiently nucleophilic exposed Cys resi-dues to react nonspecifically with the VS electrophile. Other non-DUBs include HECT or RBR-type E3 ligases and the NEDD8-specific protease SENP8, which bind Ub or Ub-like proteins and were labeled on their respective catalytic Cys residues. In contrast to IP-based methods (Supplementary Data 1), no RING-type E3 ligases or E1/E2 enzymes were detected. There is little overlap between the probe-labeled proteins and the hyper-reactive

**Fig. 6** ZUFSP is a K63-selective deubiquitinase. **a–c** Tetra-Ub chain cleavage assays to test activity and selectivity of recombinant full-length ZUFSP **a**, full-length ZUFSP purified from HEK 293T cells **b**, and recombinant catalytic domain of ZUFSP **c**. ZUFSP (1.1 μM) and Ub chains (2.2 μM) were incubated for 1 h at 25 °C, then analyzed by SDS-PAGE and silver staining. **d** Time-course analysis of ZUFSP-mediated depolymerization of K48- and K63-linked tetra-ubiquitin conjugated to a TAMRA-labeled peptide, monitored by in-gel fluorescence. Figure is representative of two independent replicates. It is noteworthy that this method detects only species bearing a TAMRA fluorophore, and not the distal fragments formed after cleavage, in contrast to silver stains that detect all protein species. **e** Surface representation of K63-linked di-Ub and K48-linked di-Ub. The color-coding shows residues exhibiting [15]N differential line broadening (DLB) in red and [13]C DLB in green. Residues were colored if the lines broadened completely (signal disappearance) or the line widths were more than doubled. These DLB effects were observed upon titration of unlabeled catalytically inactive full length ZUFSP[C360A]-His[6] to differentially labeled [15]N-proximal [13]C-distal K48- or K63-linked di-Ubiquitin. K48-linked di-Ubiquitin structure from pdb entry 2kde, K63 di-Ub structure from pdb entry 2rr9

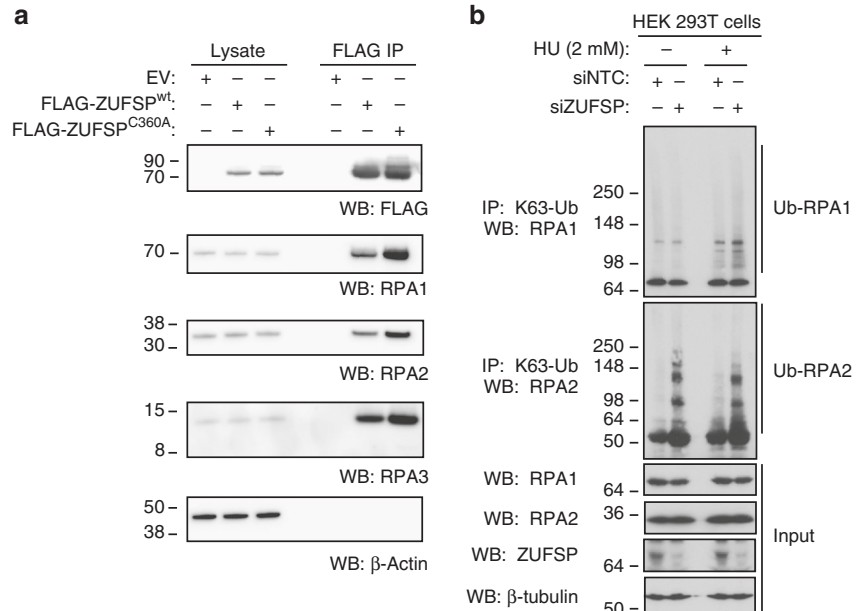

**Fig. 7** ZUFSP interacts with and affects the ubiquitination of RPA proteins. **a** FLAG-immunoprecipitated material from HEK 293T cells expressing wild-type or catalytic mutant FLAG-ZUFSP, or empty vector (EV), was analyzed by western blotting. **b** HEK 293T cells transfected with siRNA targeting ZUFSP or a non-targeting control were treated with 2 mM hydroxyurea (HU) for 2 h. K63-linked Ub chains were immunoprecipitated and the enriched proteins were analyzed by western blotting

cysteines previously identified by Weerapana et al.[61], suggesting that the VS electrophile does not react non-specifically with reactive Cys residues. Most of the labeled non-DUB proteins were not enriched by the Ub-ABP in the on-bead digestion fraction (Fig. 3b and Supplementary Data 2), perhaps indicating that only a small proportion of the protein is labeled, and that most unlabeled peptides from these proteins arise from nonspecific binding to the resin. Nonetheless, there are several enriched non-DUB proteins for which we can identify labeling sites, suggesting that some might have previously undescribed roles as Ub-binding proteins. We have shown this to be the case for ZUFSP, demonstrating that the site-specific information generated by reaction-site-centric chemoproteomics is a valuable hypothesis-generating tool. Likewise, although the SUMO2-VPS ABP does not provide site-level information about probe labeling, it strongly enriches active deSUMOylases. One protein, WDTC1, shows similar enrichment to deSUMOylases and no cross-reactivity with the Ub ABP. This protein contains a predicted SUMO interaction motif (residues 174–178[62]), suggesting it may be enriched by our method because it interacts with SUMO2.

Despite being annotated as an inactive UFM1 protease, we have demonstrated that ZUFSP (now also known as Zinc finger containing Ubiquitin Peptidase 1, ZUP1) is a catalytically active DUB and interacts with Ub-agarose, but not with UFM1-agarose. Comparison of ZUFSP with UfSP2 by sequence alignment and homology modeling suggests that ZUFSP lacks the His residue required for catalytic activity. However, UfSP1 and UfSP2 have 'atypical' catalytic triads, in which the catalytic His residue is only two amino acids away from the catalytic Asp[47,63]. In typical papain-type Cys proteases, the catalytic His and Asp are at the ends of two adjacent β-strands that form the central β-sheet (Supplementary Fig. 13). The putative catalytic His of ZUFSP, H491, appears to occupy the 'canonical' papain-like position, rather than the atypical UfSP1/2 position. However, further structural studies are required for confirmation.

ZUFSP has high selectivity for cleaving K63-linked Ub chains and a preference for long chains, with at least two distal Ub

monomers required for efficient cleavage. However, enzymatic selectivity does not arise from selective binding of K63-linked polyubiquitin, as full-length ZUFSP can interact with K48-linked polyubiquitin. Furthermore, isolated ZnFs 2, 3, and 4 interact with multiple chain types. We therefore hypothesize that the N-terminal region is important for recruitment to Ub chains, whereas the catalytic domain imparts K63 linkage selectivity. Further studies are required to understand how the ZnFs interact with Ub in the context of full-length ZUFSP, their relative binding affinities for different Ub modifications, whether other ZUFSP domains impart K63 selectivity, and whether ZUFSP is able to cleave other Ub-like proteins or mixed Ub chains. In addition, we have validated a previous observation of an interaction between ZUFSP and RPA proteins. ZUFSP depletion increases the K63 ubiquitination of RPA1 and RPA2, in particular after the induction of replication stress with HU. These data suggest that ZUFSP can exist in the same complex as RPA proteins and can regulate their ubiquitination status. Further studies are required to characterize ZUFSP substrates and to elucidate ZUFSP cellular functions.

ZUFSP requires both its putative catalytic histidine and N-terminal region for full DUB activity, but these are dispensable for probe labeling on C360. Similarly, UCHL1[C90A] is catalytically dead but nevertheless reacts substantially with Ub ABPs. The remarkable discrepancy between the probe reactivity and catalytic activity of DUBs highlights the caveats of inferring catalytic activity from ABP reactivity. Our work also demonstrates the importance of analyzing labeling sites in ABPP studies, in order to monitor off-target labeling and to identify previously undescribed enzymes.

## Methods
Synthetic procedures are reported in Supplementary Information.

**General methods and reagents**. HEK 293T and HeLa cells were obtained from Genentech's repository and authenticated following Genentech's 'Guidelines for Maintaining the Integrity of Cell Line Stocks'[64]. Cells were cultured in high-glucose Dulbecco's modified Eagle's medium (DMEM), 4.5 g/L glucose) supplemented with

10% fetal bovine serum, 1 × GlutaMax (Gibco), and 1 × Penicillin and Strepto-mycin (Gibco). Cells were maintained at 37 °C in a 10% $CO_2$ atmosphere. Samples for SDS-PAGE were heated at 85 °C for ~ 5 min in NuPAGE LDS Sample Buffer with NuPAGE Sample Reducing Agent (ThermoFisher Scientific), except for those containing purified Ub chains, which were heated at 40 °C for ~ 20 min. SDS-PAGE was performed on NuPAGE Bis-Tris Gels (ThermoFisher Scientific) with 2-(N-morpholino)ethanesulfonic acid (MES) or 3-(N-morpholino)propanesulfonic acid (MOPS) running buffer (ThermoFisher Scientific) at 200 V, or on Novex Tris-Glycine gels (ThermoFisher Scientific) with Tris-Glycine running buffer, unless otherwise stated. Silver staining was performed using the Pierce Silver Stain Kit (ThermoFisher Scientific). Coomassie staining was performed using SimplyBlue SafeStain (ThermoFisher Scientific). For western blottings, proteins were trans-ferred to nitrocellulose membranes using the iBlot dry blotting system (Thermo-Fisher Scientific), or by wet transfer (20% methanol, 115 V for 2 h) to polyvinylidene difluoride membranes, blocked in 5% non-fat milk in phosphate-buffered saline (PBS) with Tween 20 (for chemiluminescent detection) or PBS-based Odyssey Blocking Buffer (LiCor) for detection with fluorescent secondary antibodies. Chemiluminescent detection was performed with film or on an Azure Biosystems c300 western blot imaging system, whereas fluorescent detection was performed with a Li-Cor Odyssey CLx scanner. The following antibodies were used in this study: HA-horseradish peroxidase (HRP) (clone HA-7, mouse, 1:4,000, Sigma, catalog number H6533), FLAG-HRP (clone M2, mouse, 1:10,000, Sigma, catalog number A8592), UCHL1 (mouse, 1:5,000, Invitrogen, catalog number 480012), USP14 (clone D8Q6S, rabbit, 1:2,000, Cell Signaling Technologies, catalog number 11931), ZUFSP (rabbit, 1:500, Sigma, catalog number HPA044426), RPA1 (clone 4D9, rat, 1:1,000, Cell Signaling Technologies, catalog number 2198 and rabbit, 1:1,000, Cell Signaling Technologies, catalog number 2267), RPA2 (clone 4E4, rat, 1:1,000, Cell Signaling Technologies, catalog number 2208), RPA3 (rabbit, 1:1,000, Sigma, catalog number HPA005708), ATR (clone E1S3S, rabbit, 1:1,000, Cell Signaling Technologies, catalog number 13934), PARP (clone 46D11, rabbit, 1:1,000, Cell Signaling Technologies, catalog number 9652) β-actin-HRP (clone 8H10D10, mouse, 1:1,000, Cell Signaling Technologies, catalog number 12262), β-tubulin-HRP (clone 9F3, mouse, 1:10,000, Cell Signaling Technologies, catalog number 5346), α-tubulin (mouse, 1:1,000, Li-Cor, catalog number 926–42213), β-tubulin (rabbit, 1:1,000, Li-Cor, catalog number 926–42211), β-actin (rabbit, 1:1,000, Li-Cor, catalog number 926–42210). IRDye fluorescent secondary anti-bodies (Li-Cor), IRDye 800CW streptavidin (Li-Cor, catalog number 925–32230), and HRP-conjugated secondary antibodies (Jackson ImmunoResearch) were used at 1:10,000. HA-Ub-VS, HA-SUMO2VS, Ub-AMC, Ub4-Rh110, and wild-type UCHL1 were purchased from Boston Biochem. HA-Ub-PA and HA-Ub-VME were purchased from UbiQ Bio. Di- and tetra-Ub chains were purchased from Boston Biochem, except for K27-linked di-Ub, which was purchased from UBPBio.

Uncropped scans of the most important blots and gels are provided in Supplementary Figs. 22–26.

**Activity-based protein profiling**. Lysis and probe labeling: HEK 293T cells were cultured in 15 cm dishes cells until ~ 80 % confluent. The cells were collected by rinsing the plate with ice-cold PBS (2 × 10 mL) followed by scraping. Cells were transferred to 1.5 mL microcentrifuge tubes and pelleted by centrifugation at 4 °C (3 min, 350 × g). The pellet was flash-frozen in liquid nitrogen and stored at – 80 °C until lysis. Frozen cells were lysed by quickly re-thawing them in Buffer A (PBS pH 7.4, 250 mM sucrose, 5 mM $MgCl_2$) supplemented with 5 mM tris(2-carboxyethyl) phosphine (TCEP), 2 mM ATP, and 500 μM phenylmethylsulfonyl fluoride (PMSF), vortexing the cells until the pellet was suspended. The suspension was sonicated on ice (3 × 10 s bursts, 30% power, Qsonica Q55 Sonicator) and the lysate was cleared by centrifugation at 4 °C (7 min, 18,000 × g). The protein concentration determined using the Pierce BCA Protien Asasy Kit Reducing Agent Compatible (ThermoFisher Scientific) and adjusted to 5 mg/mL. Five milligrams of this cell lysate were incubated with 6.6 μg of the indicated ABP for 1 h at 25 °C, rotating at 1250 r.p.m. (Eppendorf Thermomixier). The reaction was terminated by adding 20 μL of a 20 % SDS solution, to give a final SDS concentration of 0.4 %, with continued rotation for 30 min at 25 °C. The probe-treated lysate was stored at – 80 °C before use.

Anti-HA IP: 1 mL of 5 mg/mL probe-treated lysate was diluted 10× with Buffer B (50 mM Tris-HCl pH 7.5, 150 mM NaCl, 5 mM EDTA) supplemented with 0.5% NP-40 and cOmplete Mini EDTA-free Protease Inhibitor Cocktail (Roche, Mannheim, Germany). Anti-HA affinity matrix (100 μL slurry, Roche) was added and the samples were rotated overnight at 4 °C. The next day the beads were pelleted by centrifugation at 4 °C (3 min, 2,000 × g) and the supernatant was removed. The beads were washed with ice-cold Buffer B supplemented with 0.5% NP-40 (3 × 1 mL), once with ice-cold Buffer B (1 mL) and with ice-cold 15 mM triethylammonium bicarbonate (TEAB) pH 8.0 (3 × 1 mL). To elute the immunoprecipitated material, 330 μl of 15 mM TEAB pH 8.0 (Sigma-Aldrich) containing 1 mg/mL HA peptide (ThermoFisher Scientific) and 0.02 % RapiGest SF Surfactant (Waters, Milford, MA) was added, and the samples were incubated at 37 °C for 30 min at 1,000 r.p.m. shaking in an Eppendorf Thermomixer (Eppendorf AG). Beads were pelleted by centrifugation (3 min, 2600 × g) and the supernatant was collected using a gel loading pipette and stored at – 80 °C until further processing by FASP and MS.

Filter-assisted sample preparation: Following anti-HA IP, the eluted proteins were digested with trypsin using FASP as described previously[19] with

modifications. Eluents were added to Microcon-30k filtration devices (Millipore, Billerica, MA) and briefly washed by 0.2 mL of 8 M urea in 200 mM TEAB, pH 8.5. In between each step, liquid was cleared by centrifugation at 14,000 × g, except if noted. Proteins were reduced by dithiothreitol (DTT) for 20 min at 60 °C and subsequently alkylated by iodoacetamide (IAA) for 15 min in the dark. The membrane was further washed by 8 M urea once and 200 mM TEAB three times. Trypsin (Promega, Madison, WI) was added at 1:40 enzyme/substrate ratio. Devices were briefly centrifuged at 100 × g for 30 s and incubated overnight at 37 °C. Tryptic peptides were recovered by centrifugation at 14,000 × g for 4 min. An additional 60 μL of 200 mM TEAB was added to the devices and centrifuged as an additional elution step. This eluent was combined with the previous eluent for further processing and stored at – 20 °C before clean-up and LC-MS/MS analysis.

Biotin-DADPS tagging, biotin–streptavidin affinity purification, and on-bead digestion: 50 μL Pierce Streptavidin UltraLink Resin (ThermoFisher Scientific) (100 μL of 50% slurry) was washed with PBS (3 × 0.5 mL) in a Bio-Spin spin column (Bio-Rad) with brief centrifugation. One milliliter of 5 mg/mL probe-treated lysate was added to the spin column. The column was capped and rotated gently at room temperature (rt) for 30 min. The cleared lysate was collected by brief centrifugation and the resin was washed with PBS (2 × 0.75 mL), combining washes with the lysate. Twenty-five microliters of lysate was retained for analysis. The diluted lysate was divided into two 1.25 mL aliquots in 2 mL Protein Lo-Bind tubes (Eppendorf). The Cu-catalyzed azide-alkyne cycloaddition was performed essentially as described[65]. To each tube the following reagents were added in the stated order, vortexing between the addition of each component: biotin-DADPS-azide[23] (28.3 μL of 5 mM stock in dimethyl sulfoxide (DMSO), final concentration 100 μM), freshly prepared TCEP (28.3 μL of 50 mM stock in water, neutralized with concentrated NaOH, final concentration 1 mM), tris[(1-benzyl-1H-1,2,3-triazol-4-yl)methyl]amine (TBTA) (85 μL of 1.7 mM TBTA in 4:1 tert-butanol/DMSO, final concentration 100 μM), and $CuSO_4$ (14.2 μL of 100 mM $CuSO_4$, final concentration 1 mM) or 14.2 μL water (Cu-free control). The tubes were rotated at rt for 1.5 h then quenched with EDTA (10 μL of 0.25 M solution). The two tubes were combined into a 50 mL conical tube and 25 μL was retained for analysis. Chloroform-methanol precipitation was used to remove excess biotin reagent: 10 mL ice-cold methanol and 2.5 mL chloroform were added, and the tube was briefly vortexed. Ice-cold water (7.5 mL) was added and the tube was vortexed thoroughly. The tube was centrifuged at 4 °C (10 min, 9,000 × g). The top and bottom layers were removed, rolling the tube so that the protein disk sticks to the side. Cold methanol (1.5 mL) and 375 μL chloroform were added, and the pellet was sonicated briefly (30% power, 3–5 s bursts) to resuspend the protein. Cold water (1.5 mL) was added and the tube vortexed thoroughly. The tube was centrifuged at 4 °C (10 min, 9,000 × g), and the top and bottom layers were removed. The pellet was suspended in 1.5 mL cold methanol with brief sonication, the tube was centrifuged at 4 °C (15 min, 12,000 × g), and the supernatant was discarded. The pellet was air-dried at rt and stored at – 20 °C before use.

To the pellet was added 0.8 mL 2.5% SDS/6 M freshly prepared urea in PBS. The pellet was sonicated briefly to resuspend the protein, then a further 1 mL PBS was added and sonication (30% power, 5 s pulses) was used to fully solubilize the protein. PBS (8 mL) was added to bring the SDS concentration to ~ 0.2%. Pierce Streptavidin UltraLink Resin (125 μL; ThermoFisher Scientific) (250 μL of 50% slurry) was washed with PBS (3 × 0.5 mL). The beads were resuspended in PBS (~ 1 mL) and transferred to the protein solution. The beads were rotated for 1–2 h at rt and then 4 °C overnight. Beads were pelleted by centrifugation at rt (3,000 × g, 3 min). Most supernatant was removed using a 10 mL pipette and retained for analysis. The beads were washed at rt with 3 × 1.5 mL of each of the following solutions: (i) 2% SDS, 10 mM EDTA in PBS; (ii) 1 M NaCl, 0.2% NP-40 in PBS; (iii) freshly prepared 8 M urea in PBS; (iv) PBS. For the first of each set of washes, the beads were rotated for 5–10 min at rt. Before the final wash, the beads were transferred to a Screw Cap Pierce Spin Column (ThermoFisher Scientific). The beads were resuspended in 0.5 mL freshly prepared 6 M urea in PBS. TCEP (10 μL of 500 mM in water, pH 7.0, final concentration 10 mM) was added and the tube was rotated at 37 °C for 30 min. IAA (25 μL of freshly prepared 500 mM solution in water, final concentration 25 mM) was then added and the tube was rotated at rt in the dark for 30 min. Bead suspension (12.5 μL) was retained for analysis. The tube was pulsed briefly in a centrifuge and the flow-through was discarded. The beads were washed with PBS (2 × 0.5 mL) and 50 mM Tris pH 8.0 (2 × 0.5 mL), with brief pulses in the centrifuge.

For Lys-C/trypsin digestion, the beads were resuspended in 67 μL freshly prepared 6 M urea in 50 mM Tris pH 8.0. Two microliters of 2 mg/mL Trypsin/Lys-C Mix, Mass Spec Grade (Promega) was added and the tube was rotated for 3 h at 37 °C. Three hundred anf thirty-three microliters of 50 mM Tris pH 8.0 was added to reduce the urea concentration to 1 M, and the tube was rotated overnight at 37 °C. The tube was placed in a clean 2 mL Protein LoBind Tube (Eppendorf) and pulsed briefly in a centrifuge to collect the flow-through. The beads were rinsed with 200 μL 1 M urea/0.02% RapiGest SF Surfactant (Waters) in 50 mM Tris pH 8.0, followed by 200 μL water, pulsing briefly in a centrifuge each time and combining the flow-through with the previous fraction to form the 'on-bead digestion fraction.' This was stored at – 80 °C before clean-up and LC-MS/MS analysis. The beads were subsequently washed further with 50 mM Tris pH 8.0 (3 × 500 μL) and water (3 × 500 μL). The beads were then either treated with a second protease before linker cleavage, or the linker was cleaved immediately, as described below.

Second on-bead digestion and linker cleavage: For a second on-bead digestion, beads were washed with 500 µL of 50 mM ammonium hydrogen carbonate pH 8.0, then suspended in 500 µL 50 mM ammonium hydrogen carbonate pH 8.0. Five micrograms of Glu-C or chymotrypsin (Roche, 5 µL of 1 mg/mL stock in 50 mM ammonium hydrogen carbonate pH 8.0) was added and the tube was rotated at 25 °C overnight. The tube was pulsed briefly in a centrifuge and the flow-through was discarded. The beads were washed with 50 mM ammonium hydrogen carbonate pH 8.0 (3 × 500 µL), then water (3 × 500 µL), discarding the flow-through. For all samples, the DADPS linker was cleaved by adding 400 µL of 2% formic acid (FA) in water, rotating the tube at rt for 1 h. The tube was pulsed briefly to collect the supernatant and the cleavage step was repeated with 200 µL of 2% FA in water (30 min rotation). The beads washed with 50% acetonitrile (ACN)/water (2 × 200 µL) and all four eluents were combined to form the 'cleaved fraction.' Peptides were dried on a SpeedVac (Savant) and stored at – 20 °C before clean-up and LC-MS/MS analysis.

**LC-MS/MS analysis.** For data presented in Figs. 1b, 2b, and Supplementary Tables 2, 3, and 4: For a protein input of 5 mg, 10% of the peptide sample was dried in a SpeedVac and de-salted with a C18 STAGE tip (Proxeon, Thermo Fisher). Peptides were loaded onto an Acquity UPLC BEH130 C18 column (1.7 µm, 130 Å, 100 µm × 100 mm) at a flow rate of 1.5 µl min$^{-1}$ in solvent A (98% water/2% ACN/ 0.1% FA) using a NanoAcquity UPLC system (Waters). Separation was achieved at 1.0 µl min$^{-1}$ with a linear gradient ranging from 2% solvent B (98% ACN/2% water/0.1% FA) to 25% solvent B over 45 min. Eluted peptides were injected onto an Orbitrap Elite mass spectrometer (Thermo Fisher) using an Advance Captive-Spray source (Michrom) at a voltage of 1.2 kV. Full MS scans were collected in the Orbitrap at 60,000 resolution. The top 15 most abundant precursor ions were selected for data-dependent fragmentation via collision-induced dissociation (CID) and MS/MS collected in the ion trap.

For data presented in Figs. 3, 4a, and 5b, Table 1, Supplementary Table 5, Supplementary Figs. 5–9, and Supplementary Data 1–4: From a protein input of 5 mg, 10% (from FASP or on-bead digestion), or 60% (of the cleaved fraction, following linker cleavage) of the sample was dried in a SpeedVac and de-salted with a C18 STAGE tip (Proxeon, Thermo Fisher). Peptides were loaded onto a New Objective PicoFrit Acquity BEH130Å C18 column (1.7 µM, 100 µM × 250 mm) with a flow rate of 0.8 µL/min in Solvent A (98% water/2% ACN/0.1% FA) using a NanoAcquity UPLC system (Waters). Separation was achieved at a flow rate of 0.5 µL/min with a linear gradient of 2% solvent B (98% ACN/2% water/0.1% FA) to 35% solvent B over 40 min. Eluted peptides were injected onto an Orbitrap Fusion Tribrid mass spectrometer (Thermo Fisher) using a Nanospray Flex Ion-Source (Thermo Scientific) at a voltage of 1.9 kV. Full MS scans were collected in the Orbitrap at 120,000 resolution. Data-dependent MS2 scans were collected in Top Speed mode using a cycle time of 3 s between Full MS scans. Fragmentation was achieved with higher-energy collisional dissociation activation type using 25% collision energy while spectra were collected in the ion trap.

For the experiments presented in Supplementary Table 5 and Supplementary Fig. 7, a protein input of 1 mg was used and 50% of the sample was injected.

Searches (for all ABPP data sets): MS/MS spectra were searched using Mascot (version 2.3.02) against a database comprising the human proteome (Uniprot; December 2015), known contaminants (i.e., proteolytic enzymes), along with all decoy sequences. Search parameters included trypsin, trypsin + Glu-C or trypsin + chymotrypsin cleavage with up to two missed cleavage events, precursor ion tolerance of 50 p.p.m. and fragment ion tolerance of 0.8 Da. Searches also permitted variable modifications of methionine oxidation (+ 15.9949 Da) and either cysteine modification by carbamidomethylation (+ 57.0215) or the mass of the relevant reacted DUB probe remnants (VS: + 192.0569; VME: + 172.0848; PA: + 112.0637; VPS (DADPS linker): + 387.1940; VPS (DDE linker): + 344.1631; hydrolyzed VPE probe: + 158.0691). Peptide spectral matches were filtered to do linear discriminant analysis to a false discovery rate (FDR) of 5% on the peptide level and subsequently at 2% on the protein level for Orbitrap Elite acquired sample sets. For Orbitrap Fusion Tribrid samples, a FDR of 1% on the peptide level was applied, with additional post-hoc filtering for ions scores greater than 10 and restrictions of ± 5 p.p.m. mass accuracy. The Ascore algorithm was used to confirm site localization peptides bearing probe specific modifications and more than one Cys residue[66].

Analysis of protein enrichment with Ub-VPS probe relative to negative controls (SUMO2-VPS or omission of copper): For each protein, the number of features (i.e., unique peptide forms with a particular charge state) detected in each of three replicates across the two conditions (Ub-VPS vs. control) was determined. Proteins with fewer than a total of three features across all samples and replicates were discarded. These triples of feature counts were then compared between the two conditions by the voom method commonly applied in RNA-Seq analysis[67]. This yields, for each protein, a log$_2$ fold change in the number of features found by the two probes paired with an FDR-adjusted p-value of the hypothesis that the two probes have equal affinity for the protein (i.e., that the protein binds non-specifically). Computations were performed in R version 3.4.3, using the R Bioconductor package limma[68], which includes the *voom* method.

**Expression and purification of UCHL1 mutants.** The DNA sequence for full-length UCHL1$^{C90A}$, UCHL1$^{C152A}$, and UCHL1$^{C90A, C152A}$ was synthesized by GenScript and cloned into pET3a expression vectors.

Plasmids were introduced into *E. coli* BL21 A.I. cells (ThermoFisher, San Jose, CA) to produce full-length UCHL1$^{C90A}$, UCHL1$^{C152A}$, or UCHL1$^{C90A,C152A}$. Recombinant protein expression was induced by the addition of isopropyl β-D-1-thiogalactopyranoside (IPTG) and arabinose to the cultures, along with continued shaking at 200 r.p.m. for 4 h at 37 °C. Resulting cell pellets were resuspended in a buffer containing 50 mM HEPES pH 8 and 1 mM EDTA, and lysed by microfluidization. Overexpressed proteins were purified by two rounds of ion exchange chromatography (IEX) using a Q FF column (GE Lifesciences) with a gradient of elution buffer containing 50 mM HEPES pH 8, 1 M NaCl, and 1 mM EDTA. Between rounds of purification, the NaCl concentration was reduced by dilution. Protein purity was determined by SDS-PAGE analysis and densitometry. Activity of the purified UCHL1 mutants was analyzed by a standard fluorescence assay using Ub-AMC (Boston Biochem) as a substrate.

**UHCL1 probe labeling.** Recombinant UCHL1 (2 µM) was reacted with 1 or 5 eq Ub-VPS in a total volume of 40 µL of Buffer A supplemented with 5 mM TCEP. The reaction was allowed to proceed for 1 h at 25 °C with rotation (800 r.p.m.) in a Thermomixer (Eppendorf). The reaction was quenched with 21 µL of 2.5:1 NuPAGE LDS Sample Buffer:NuPAGE Sample Reducing Agent (ThermoFisher Scientific), and heated at 85 °C for 5 min. Fifteen microliters of sample was resolved by SDS-PAGE, using a 10% Bis-Tris gel with MES running buffer. Proteins were detected using SimplyBlue SafeStain (ThermoFisher Scientific).

**Ub-AMC hydrolysis assay.** Ten microliters of 20 nM UCHL1 (final concentration 10 nM) or 500 nM ZUFSP-His$_6$ (final concentration 250 nM) was added to 10 µL of 500 nM Ub-AMC (Boston Biochem; final concentration 250 nM) in PBS buffer (pH 7.4) containing 5 mM TCEP and 0.1 mg/mL bovine serum albumin (BSA) in a black ProxiPlate-384 Plus F 384-shallow well microplate (PerkinElmer). Fluorescence was detected on a Wallac EnVision 2104 Multilabel reader (PerkinElmer; excitation 355 nm, emission 460 nm) with reads interspaced by 10 s of shaking. The fluorescence reading from a negative control (Ub-AMC only) was subtracted from each value.

**ZUFSP construct generation.** The wild-type full-length *ZUFSP* gene (Met1-Pro578) as well as the C-terminal region (His298–Pro578) were subcloned into a modified pRK5 vector behind a cytomegalovirus promoter with N-terminal and C-terminal Flag-tags, respectively. In addition, the full-length *ZUFSP* gene and the C terminus (H298-P578) were ligated into a dual expression vector containing both a polyhedrin promoter for insect cell expression and a T7 promoter for protein production in *E. coli*. All dual promoter constructs contained a non-cleavable C-terminal hexa-histidine tag. The individual point mutations C360A or H491A were introduced by site-directed mutagenesis (QuikChange, Agilent Technology). The integrity of all expression constructs was confirmed by DNA sequencing.

**Pull-down with Ub-, SUMO2-, or UFM1-agarose.** HEK 293T cells (3 × 10$^6$ cells/ dish) were plated in 10 cm dishes in antibiotic-free DMEM and cultured overnight. The following day, the cells were transfected with 5 µg of the plasmid encoding FLAG-ZUFSP (full-length or residues 298–578) using 15 µL FuGENE 6 Transfection Reagent (Promega) and 0.5 mL Gibco Opti-MEM Reduced Serum Medium (ThermoFisher Scientific). After 2 days, cells were lysed and treated with probe as described above (for ABPP), using 2 × 5 mL PBS for washing and 0.5 mL supplemented Buffer A. Two hundred microliters of lysate (protein concentration 1 mg/ mL) was incubated with 20 µL (40 µL of 50% slurry) of Ub-, SUMO2- or UFM1-conjugated agarose (Boston Biochem), pre-washed with PBS (3 × 500 µL). The suspension was rotated at 4 °C for 3 h, and then centrifuged at 4 °C (3 min, 2,000 × g). The supernatant was discarded and the beads were washed at 4 °C with Buffer A supplemented with 0.5% NP-40 (3 × 0.5 mL), then with PBS (0.5 mL). Proteins were eluted from the beads by boiling in 40 µL 1 × reducing LDS sample buffer (100 °C, 7 min). For western blot analysis, 10 µg protein lysate (input) or 7 µL eluent from beads was resolved on a 4–12% Bis-Tris gel with MOPS running buffer. Proteins were transferred to a nitrocellulose membrane, probed with anti FLAG-HRP antibody, and detected by chemiluminescence.

**FLAG-ZUFSP IP from HEK 293T cells.** HEK 293T cells (10$^7$ cells/dish) were plated in 15 cm dishes in antibiotic-free DMEM and cultured overnight. The following day, 10 dishes of cells were transfected, each with 12 µg of the plasmid (encoding FLAG-ZUFSP, FLAG-ZUFSP$^{C360A}$, FLAG-ZUFSP$^{H419A}$, or an empty vector) using 36 µL FuGENE 6 Transfection Reagent (Promega) and 1.2 mL Gibco Opti-MEM Reduced Serum Medium (ThermoFisher Scientific). After 2 days, the medium was removed, cells were washed with ice-cold PBS (10 mL), and collected by scraping. The cells were centrifuged at 4 °C (3 min, 700 × g) and the supernatant was discarded. The cells were suspended in 12 mL ice-cold hypotonic lysis buffer (10 mM Tris pH 7.5, 10 mM NaCl, 1.5 mM MgCl$_2$, 0.5 mM DTT, 1 mM PMSF) and homogenized with ~ 50 strokes of a tight Dounce homogenizer. The crude lysate was centrifuged at 4 °C (5 min, 20,000 × g), the supernatant was transferred to clean tubes and centrifuged again at 4 °C (5 min, 20,000 × g). The clarified lysate was transferred to a 15 mL conical tube and the following were added to adjust the ionic strength: 110 µL 1 M Tris pH 8.0 (final overall concentration ~ 20 mM), 110 µL 0.1 M MgCl$_2$ (final overall concentration ~ 2.5 mM) and 330 µL 3 M NaCl (final overall concentration ~ 100 mM). Anti-FLAG M2 Affinity Gel slurry (500 µL;

Sigma) was washed with 10 mM glycine pH 2.5 (1 mL) and Tris-buffered saline (TBS, 50 mM Tris, 150 mM NaCl pH 7.5) (2 × 1 mL) at 4 °C, then transferred to the lysate. The tube was rotated for 4 h at 4 °C and then centrifuged at 4 °C (5 min, 1,200 × g) to pellet the beads. The supernatant was discarded and the beads were washed in 12 mL of the following buffers, with centrifugation at 4 °C (5 min, 1,200 × g) to pellet the beads between washes: once with wash buffer 1 (20 mM HEPES pH 7.9, 420 mM NaCl, 1.5 mM MgCl$_2$, 25% glycerol) and four times with wash buffer 2 (20 mM Tris pH 7.5, 20% glycerol, 300 mM NaCl, 0.1% NP-40). For the final wash, the beads were rotated overnight at 4 °C in wash buffer 2 supplemented with 0.1 mM PMSF before centrifugation. The beads were then resuspended in 1.2 mL TBS and transferred to a 1.5 mL microcentrifuge tube, then pelleted by centrifugation at 4 °C (5 min, 1,200 × g). The supernatant was removed and the beads were resuspended in 1.2 mL TBS containing 500 μg/mL 3 × FLAG peptide (Sigma). The tube was rotated at 4 °C for 4 h and the beads were separated using an empty Micro Bio-Spin column (Bio-Rad). The eluent was exchanged into 50 mM HEPES pH 7.5, 100 mM NaCl, 5% glycerol, 1 mM TCEP using a 5 mL Zeba Spin Desalting Column (5 mL, 7 K molecular weight cut-off, ThermoFisher Scientific). Proteins were analyzed by SDS-PAGE and detected using SimplyBlue SafeStain (ThermoFisher Scientific), by western blotting, or by MS (see below).

**ZUFSP-His$_6$ purification from *E. coli*.** Constructs for the expression of full-length ZUFSP (M1-P578) and C-terminal region (H298-P578), and their corresponding C360A mutants, were generated as described above. The constructs were transformed into BL21 (DE3) *E. coli* competent cells. Single-colony overnight starter cultures grown in lysogeny broth were used for inoculation of production cultures in Instant Terrific Broth overnight autoinduction medium (Novagen) and grown at 16 °C for 2 days.

After expression, cells were collected by centrifugation and bacterial paste was resuspended in lysis buffer containing 20 mM Tris pH 8.0, 500 mM NaCl, 10% glycerol, 10 mM imidazole, 2.8 mM β-mercaptoethanol, 0.1 mM PMSF, and EDTA-free protease inhibitor tablets (Roche). Cells were lysed by homogenization and microfluidization, and lysates were clarified by centrifugation at 30,000 × g for 1 h. Cleared lysates were loaded onto nickel-NTA resin (Qiagen), washed with 10 column volumes of lysis buffer (described above) supplemented with 25 mM imidazole, and eluted with lysis buffer supplemented with 250 mM imidazole. To avoid precipitation, the Ni-eluate was not concentrated but instead loaded directly onto a size exclusion column (SEC), HiLoad 16/60 Superdex 200 prep-grade column (GE Healthcare), equilibrated with PBS and 0.1 mM TCEP. For the catalytic domain constructs, Ni-NTA and SEC purification steps were sufficient to generate homogeneous and high-purity proteins. On the other hand, full-length ZUFSP required further purification; pooled SEC fractions were concentrated and purified by anion exchange using HiTrap Q HP column (GE Healthcare). Low-salt (20 mM NaCl) and high-salt (1 M NaCl) buffers supplemented with 20 mM Tris pH 8.0 and 0.1 mM TCEP were used to generate a steady salt gradient, where full-length ZUFSP was eluted at around ~ 125–150 mM NaCl. Purity was further assessed by SDS-PAGE stained with Coomassie Blue. For NMR studies, proteins were concentrated to ~ 1 mM.

**Probe labeling of purified ZUFSP.** ZUFSP (1.5 μM) was incubated with 0, 1, or 5 eq ABP (Ub-VPS, HA-Ub-VS, HA-Ub-VME, HA-Ub-PA, or HA-SUMO2VS) in PBS supplemented with 1 mM TCEP in a total volume of 20 μL. The reaction was allowed to proceed at for 1 h at 25 °C with rotation (800 r.p.m.) in a Thermomixer (Eppendorf). The reaction was quenched with 11 μL of 2.5:1 NuPAGE LDS Sample Buffer:NuPAGE Sample Reducing Agent (ThermoFisher Scientific) and heated at 40 °C for 20 min. Sample (15 μL) was resolved by SDS-PAGE, using a 4–12% Bis-Tris gel with MOPS running buffer for full-length ZUFSP and a 10% Bis-Tris gel with MES running buffer for ZUFSP(298–578). Proteins were detected using SimplyBlue SafeStain (ThermoFisher Scientific).

**Ub chain cleavage assays using purified ZUFSP.** ZUFSP (1.1 μM) was incubated with 2.2 μM Ub$_2$ or Ub$_4$ chains in a total volume of 10 μL PBS pH 7.4 supplemented with 1 mM TCEP. The reaction was allowed to proceed for the indicated time at 25 °C with rotation (800 r.p.m.) in a Thermomixer (Eppendorf). The reaction was quenched with 6 μL of 2.5:1 NuPAGE LDS Sample Buffer:NuPAGE Sample Reducing Agent (ThermoFisher Scientific) and heated at 40 °C for 20 min. The sample was diluted with 20 μL 1 × reducing LDS sample buffer and 10 μL sample was resolved by SDS-PAGE using a 10% Bis-Tris gel with MES running buffer. Proteins were detected using Pierce Silver Stain Kit (ThermoFisher Scientific).

**Generation of Ub$_4$-peptide-TAMRA conjugates.** K63- and K48-linked tetra-Ub chains were obtained from Boston Biochem Cambridge (K63-linked (catalog number UC-310B) and K48-linked (catalog number UC-210B)). The 5-TAMRA (5-Carboxytetramethylrhodamine) peptide was generated by CPC-Scientific consisting of the sequence 5-TAMRA-YPYDVPDYAIREIVSRNKRRYQEDG. K63 or K48 tetra-Ub chains were conjugated to the peptide via its lysine residue as follows: (1) generation of tetra-Ub-MESNA; incubating 250 nM E1, 10 mM MgCl$_2$, 10 mM MgATP, 1 mM tetra-Ub, 100 mM MESNA (Sigma-Aldrich, catalog number 63705), in 20 mM Na$_2$HPO$_4$ at pH 8.0 at 37 °C overnight. Dialyzed into 0.4%

trifluoroacetic acid (TFA) and tetra-Ub-MESNA was lyophilized. (2) Lyophilized tetra-Ub was dissolved in DMSO at a concentration of 0.5 mM and 2 mg of peptide were added until all components were dissolved, reaction volume 1 mL. The reaction was initiated by adding (final concentrations) *N*-hydroxysuccinimide (27.5 mM), AgNO$_3$ (3.3 mM), and 22 μl *N*,*N*-diisopropylethylamine and incubated at RT overnight. The reaction was diluted 10 × with ddH2O and desalted into PBS pH 7.5. Non-conjugated peptide was removed by SEC chromatography. In a second step, non-conjugated tetra-Ub was removed by IEX chromatography as described above and buffer exchanged into 1 × PBS (pH 7.5). The concentration of the purified final conjugate was determined by absorbance using an extinction coefficient for 5-TAMRA at 80,000 cm$^{-1}$ M$^{-1}$.

**Ub$_4$-peptide-TAMRA depolymerization studies.** For depolymerization assays, 1 μM ZUFSP-His$_6$, 1 μM ZUFSP$^{C360A}$-His$_6$, 1 μM ZUFSP(298–578)-His$_6$, and 1 μM ZUFSP(298–578)$^{C360A}$-His$_6$ were diluted in PBS buffer (pH 7.5) containing 5 mM DTT, to generate 10 × stock solutions in respect to the final concentration and pre-incubated at rt for 10 min. In a 90 μL reaction, 9 μg (2.7 μM) of 5-TAMRA-peptide/tetra-Ub (K63- or K48-linked) was mixed with 9 μL of diluted enzyme in PBS buffer (pH 7.5). Aliquots of 10 μL of the reaction were mixed with 4 μL 2 × SDS loading buffer at the time points indicated, to stop the reaction. Samples (14 μL) were subjected to SDS gel electrophoresis using precast BioRad Criterion TGX AnykD gels (catalog number 5671124). Fluorescence was analyzed using the FluorChem imager from Protein Simple according to the user manual.

**ZUFSP kinetic analysis.** Michaelis–Menten kinetic measurements with full-length ZUFSP were carried out using 200 nM ZUFSP with a series of K63-linked tetra-Ub-Rh110 substrate titrations with at least three technical replicates. Samples were reacted in a buffer consisting of 50 mM HEPES (pH 7.5), 100 mM NaCl, 2.5 mM DTT, 0.1% (w/v) bovine γ globulin (Sigma, catalog number G5009–25G). The starting substrate concentration of K63-linked Tetra-Ub-Rho110 (Boston Biochem, catalog number UC-355) used for the Michaelis–Menten analysis was 25 μM down to 625 nM. Reactions were carried out for 1 h at rt in a black 100 μL volume 96-well half-area plates (Corning, catalog number 3993). The enzymatic activity was calculated by fitting the data using the initial velocity using the linear $V_0$ values measured by analyzing the fluorescence signal of cleaved Rho-110 using excitation at 485 nm and emission at 535 nm.

**Generation of isotopically labeled di-Ubs.** For both K48- and K63-linked di-Ub, the distal ubiquitin is $^{15}$N labeled, whereas the proximal Ub is $^{13}$C labeled.

Ub was cloned into a peT3a vector and transformed into BL21-CodonPlus (DE3) strain (Agilent, catalog number 230280) and expressed with the following modifications: bacterial cultures were grown at 37 °C in M9 media supplemented with either $^{15}$NH$_4$Cl (2 g/L) (Cambridge Isoptopics, catalog number NLM-467-PK) or $^{13}$C$_6$-glucose (4 g/L) (Cambridge Isoptopics, catalog number CLM-1396-PK). After reaching an optical density of 0.6, the cells were cooled to 30 °C, supplemented with additional $^{15}$NH$_4$Cl (1 g/L) or $^{13}$C$_6$-glucose (2 g/L), induced with 1 mM IPTG and further grown for 10 h. Protein purification was performed at rt. Cells were collected and lysed in lysis buffer (50 mM HEPES 7.0). The cleared lysate was subjected to affinity chromatography using DEAE sepharose Fast Flow (GE, catalog number 17-0709-01). Ub was collected in the flow through and dialyzed overnight into NaOAc (pH 4.5). Dialyzed material was clarified by centrifugation at 35 K and Ub was subjected to IEX using a MonoS column (GE Healthcare, catalog number 17-5169-01). The following enzymes were obtained from Boston Biochem (Cambridge): UBE1 (catalog number E-305), UBE2K, and UBE2N/UBE2V1 complex (catalog numbers E2-602 and E2-664), respectively. K63- and K48-linked di-Ub chains were generated and purified as follows: in separate reactions incubating 250 nM E1 enzyme, 5 μM UBE2K (K48 linked) or 5 μM UBE2N/UBE2V1 complex (K63 linked) with equal molar ratios of differentially labeled Ubs (each at 1 mM), 10 mM ATP, 50 mM HEPES pH 8.0, 10 mM MgCl$_2$ in a 10 mL reaction at 37 °C. After 2 h, the reaction was acidified with 2 mL of 17.4 M Glacial Acetic Acid. Obtained di-Ubs (K63- or K48-linked) were purified by cation exchange using a MonoS column (GE Healthcare, catalog number 17-5169-01). All purified di-Ub chains were buffer exchanged into 1 × PBS buffer and proteins were flash frozen in liquid nitrogen before storage at − 80 °C.

**Cloning and expression of ZUFSP ZnFs.** The ZUFSP ZnF1 (M1-E28), ZnF2 (M25-R60), ZnF3 (E148-P184), and ZnF4 (D189-F221) domains were cloned into the EitrMBP vector with N-terminal His6-MBP tag and TEV cleavage site and further transformed into BL2 (DE3) *E. coli* strain. Expression of $^{15}$N-labeled proteins for NMR studies was carried out in M9 media at 16 °C for ~ 20 h using 0.2 mM IPTG induction.

The purification of all proteins was carried out at 4 °C using Ni-NTA resin (Qiagen), followed by protease cleavage and another Ni$^{2+}$ affinity chromatography to remove the purification tag. Proteins were further purified by phenyl column followed by SEC chromatography (Superdex 75) using a buffer consisting of 50 mM Tris pH 8.0, 100 mM NaCl, and 0.5 mM TCEP.

**Protein NMR methods.** For labeled di-Ub with unlabeled ZUFSP: NMR spectra were recorded on a Bruker 800 MHz spectrometer operating at 18.8 Tesla using

triple resonance cryogenic probes optimized for proton detection. All two-dimensional spectra were acquired with a spectral width of 12 p.p.m., and 1,270 data points in the direct proton dimension and 41 p.p.m. and 256 sample points in the $^{15}$N dimension with States-TPPI (time proportional phase incrementation) type selection. The resulting free induction decay resolution was 15.14 and 26.02 Hz/data point, respectively. All spectra were recorded at 300 K. The pH was adjusted to 7.2 without correction for isotope shifts. For data processing the BRUKER software package TOPSPIN 3.5pl6 was used. The signal intensities, volumes, and line widths were automatically calculated in CCPN[69]. Visualization and presentation of the three-dimensional tertiary di-Ub structures from the RSCB Protein Database was done in Pymol (The PyMOL Molecular Graphics System, Version 1.7.4 Schrödinger, LLC). The sequential assignment of Ub is deposited in the Biological Magnetic Resonance Bank and can be retrieved with the accession number 4769. All samples contained 137 mM NaCl, 10 mM $Na_2HPO_4$, 27 mM KCl, and 1.8 mM $KH_2PO_4$ adjusted to a pH of 7.2 and contained 7% (w/w) $D_2O$. ZUFSP titration experiments were done by addition of unlabeled, catalytically inactive, full-length ZUFSP$^{C360A}$-His$_6$ to differentially labeled $^{15}$N-proximal $^{13}$C-distal di-Ub. The di-Ubs were at 60 μM concentration and the ZUFSP added from a stock solution of 500 μM to a ratio of 1:1. All proton chemical shifts were referenced to internal dextran sodium sulfate (DSS) (50 μM) and $^{15}$N referenced indirectly using the $^1$H chemical shift of the methyl group in DSS by multiplication with a factor of 0.101329118.

For labeled ZnFs with unlabeled Ub: NMR experiments were performed at 298 K on a Bruker 500 MHz spectrometer. All NMR samples were prepared in buffer containing 50 mM Tris (pH 8.0), 100 mM NaCl, 0.5 mM TCEP, and 10% (v/v) $D_2O$ with concentration of ~ 0.1 mM. Titration experiments were performed on $^{15}$N-labeled ZUFSP ZnFs by adding unlabeled mono- and di-Ub (K48, K63 and linear) to a ratio of 1:1. The data were analyzed using Sparky and NMRViewJ software.

**Competition experiments with Ub chains.** A three-fold dilution series of mono-Ub, K48-linked di-Ub, K48-linked tetra-Ub, or linear tetra-Ub was premixed with ZUFSP-His$_6$ (final concentration 100 nM). K63-linked Ub$_4$-Rh110 (final concentration 750 nM) was added to initiate the reaction and the plate was centrifuged briefly. Assays were performed in technical triplicate in a total volume of 20 μL of 20 mM Tris pH 7.5, 150 mM NaCl, 1 mM DTT, and 1 mg/mL BSA, in a black ProxiPlate-384 Plus F 384-shallow-well microplate (PerkinElmer, catalog number 6008260). Fluorescence was detected on a Wallac EnVision 2104 Multilabel reader (PerkinElmer; excitation 485 nm, emission 535 nm) with reads interspaced by 30 s of shaking. Initial rates (RFU/min) were calculated from the linear increase in Rh110 fluorescence and expressed as a percentage of the maximum rate (without additional Ub chains).

**Affinity purification–MS for FLAG-ZUFSP.** SDS-PAGE and in-gel digestion: 10 μg of FLAG-ZUFSP, 10 μg of FLAG-ZUFSP$^{C360A}$, or a similar volume of eluent from an empty vector IP (immunoprecipitated as described above) were resolved on 4–12% Bis-Tris gels using 1 × MOPS running buffer at 200 V. The SDS-PAGE gel was stained with SimplyBlue SafeStain (ThermoFisher Scientific) and bands excised in 10 gel regions from the 10 kDa mass up to the loading edge of the gel, for each sample. Gel bands were washed with 50 mM ammonium bicarbonate (AMBIC)/50% ACN solution for 15 min with mild shaking. They were further dehydrated with 100% ACN and mild shaking for an additional 5 min with subsequent speedvac drying. Gel bands were rehydrated with a solution of trypsin (Promega) at a concentration of 0.02 μg/μL in 25 mM AMBIC and incubated on ice for at least 1 h. Excess trypsin solution was removed before covering gel bands with 25 mM AMBIC and incubated overnight at 37 °C. Peptides were extracted with 100 μl of 10% ACN/0.1% TFA with mild shaking for 15 min. After completely drying down samples, they were reconstituted with 2% ACN/0.1% FA before injection onto the mass spectrometer.

LC-MS/MS methods: Peptides were loaded onto an Acuity UPLC BEH130 C18 column (1.7 μm, 12 Å, 100 μm × 100 mm) at a flow rate of 1.5 μL/min using a NanoAcquity UPLC system (Waters). Separation was achieved with a linear gradient of 2% Solvent B to 25% Solvent B over 60 min with a total method run time of 60 min. Solvent B is 98% MeCN/2% water/0.1% FA, whereas Solvent A consists of 98% water/2% MeCN/0.1% FA. Eluted peptides were injected onto an Orbitrap Elite mass spectrometer (ThermoFisher) using an Advance CaptiveSpray source (Bruker, Auburn, CA) at a voltage of 1.3 kV. Full ms scans were collected in the orbitrap at 60,000 resolution and the top 15 most abundant ions were selected in a data-dependent mode and fragmented with CID in the ion trap.

MS/MS spectra were searched using Mascot (v.2.3.02) against a human proteome database along with a targeted decoy database (Uniprot April. 2015) with known contaminants. Search parameters included trypsin cleavage allowing up to two missed cleavage events, a precursor ion tolerance of 50 p.p.m., and a fragment ion tolerance of 0.8 Da. Searches also permitted variable modifications of lysine ubiquitination (+ 114.0429 Da), methionine oxidation (+ 15.9949 Da), and cysteine carbamidomethylation of (+ 57.0215). Peptide spectra matches were filtered with a FDR of 5% on the peptide level and subsequently at 2% on the protein level.

**ZUFSP knockdown and IP of K63-linked Ub chains.** Cells were plated in 15 cm dishes (HEK 293T: $7.5 \times 10^6$ cells/dish; HeLa: $2.5 \times 10^6$ cells/dish) in 20 mL

antibiotic-free high-glucose DMEM, and allowed to adhere for 4–6 h. They were then transfected with ZUFSP small interfering RNA (siRNA) (Silencer Select, ThermoFisher Scientific, catalog number 4392420, assay ID s48009) or negative control (Silencer Select Negative Control No. 1 siRNA, ThermoFisher Scientific, catalog number 4390843) at a final concentration of 10 nM, using 34 μL Lipofectamine RNAiMAX (ThermoFisher Scientific) and 1 mL Opti-MEM Reduced Serum Medium (ThermoFisher Scientific) per dish. After 12–24 h, the medium was removed and replaced with 25 mL antibiotic-containing high-glucose DMEM. After a further 3 days (total 4 days knockdown), HEK 293T cells were treated with 2 mM HU for 2 h, or left untreated. HeLa cells were trypsinized, re-plated and transfected as before, and allowed to grow for a further 3 days before being treated with 2 mM HU for 2 h, or left untreated. Two plates of cells were used per condition. The medium was removed and the cells were washed twice with PBS. Urea lysis buffer (750 μL; 20 mM Tris pH 8.0, 2.5 mM $MgCl_2$, 100 mM NaCl, 6 M urea, 0.1% Triton X-100, 10 mM N-ethyl maleimide (NEM), complete protease inhibitor tablets (Roche), phosphatase inhibitor cocktail-1 and -3 (Sigma), and 25 μM MG-132) was added per plate, and the cells were collected by scraping. The cells were centrifuged at $18,000 \times g$ at 4 °C for 10 min and the supernatant was removed. Pellets were sonicated with an additional two pellet volumes of urea lysis buffer, centrifuged at $18,000 \times g$ at 4 °C for 10 min, and the supernatants were pooled. Lysate was pre-cleared with 30 μL protein A beads (Sigma) and 2.5 μg non-production grade Trastuzumab per IP for 1.5–3 h. Protein inputs were equalized and lysate was diluted ~ 2-fold using Ub chain IP lysis buffer (20 mM Tris pH 7.5, 135 mM NaCl, 1.5 mM $MgCl_2$, 1% Triton X-100, 1 mM EGTA, 10% glycerol, 50 mM NaF, 10 mM NEM, complete protease inhibitor tablets (Roche), phosphatase inhibitor cocktail-1 and -3 (Sigma), 25 μM MG-132 and 4 M urea) to give equal volumes per IP. Pre-cleared lysates were subsequently immunoprecipitated with 2 μg anti-K63 antibody per mg total protein overnight at 4 °C, and immunocomplexes were captured with 25 μl Protein A beads (Sigma) per IP for 2 h. Beads were washed 4 × with Ub chain IP lysis buffer without urea and proteins were eluted by boiling in reducing sample buffer.

**Statistical analysis.** For ABPP studies employing a negative control, experiments were performed in biological triplicate, the largest number of replicates tenable given available resources. For analysis, moderated $t$-tests were employed, comparing feature counts for each protein between probe and control. This was performed using the LIMMA-VOOM algorithm[67], which is designed to make large numbers of simultaneous comparisons between groups with small sample sizes. The LIMMA-VOOM algorithm first estimates within-sample variation across a small number of replicates for each condition and protein feature, then applies an empirical Bayes strategy to borrow strength across features. These empirical Bayes estimates of variances are applied in conducting the moderated $t$-tests, which in turn yield $p$-values displayed in the volcano plot. Variances for counts generated from the two probes for each protein were derived from the empirical Bayes procedure underlying LIMMA and are incorporated directly into the moderated $t$-tests of its algorithm in accordance with the published workflow so as to preserve the performance of statistical procedures, as previously described[68].

**Data availability.** All MS raw datasets and peptide-spectrum matches have been deposited to the MASSIVE database (http://massive.ucsd.edu/) and can be downloaded by the identifier MSV000081972. The data that support the findings of this study are available from the corresponding author upon request.

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

## Acknowledgements

We thank Yvonne Franke and the BioMolecular Resources team for construct design and protein expression trials, Steve Staben for advice on probe design and synthesis, Si-Han Chen for evaluating ZUFSP siRNA and antibodies, and Matthew Chang for analyzing RNA-Seq data. We thank Dhara Amin, Willem den Besten, Karl Merrick, and Wayne Fairbrother for helpful discussion. We also thank Don Kirkpatrick and Jennie Lill for critical reading of the manuscript and for advice on proteomics methods. M.B. acknowledges the support of NIH grant R01 GM111703.

## Author contributions

D.S.H., J.H., and I.E.W. designed ABPs. D.S.H. and J.H. performed ABPP experiments. D.S.H. performed chemical synthesis and in vitro biochemical experiments. T.P.M. and K.Y. performed and analyzed LC-MS/MS experiments. F.E.O. and A.A. developed the total linear chemical synthesis of the SUMO2 protein. F.E.O. synthesized the ABPs. G.T. C., B.B., and C.S. expressed UCHL1 proteins, and synthesized and performed experiments with tetra-Ub-modified fluorescent peptides. W.F.F. performed statistical analysis. A.P.A. expressed ZUFSP protein. T.M. performed protein NMR experiments. N.P. expressed ZUFSP ZnFs and performed protein NMR experiments. D.S.H., J. H., I.E.W., T.P., J.F., D.K., and M.B. designed experiments and analyzed data. D.S.H. and I.E.W. wrote the manuscript with input from all authors.

## Additional information

**Competing interests:** D.S.H., T.P.M., A.P.A., D.K., T.P., N.P, T.M., W.F.F., K.Y., and I.E. W. are current employees of Genentech Inc. J.H. and J.F. are former employees of Genentech Inc. J.F. is a current employee of Merck Inc. F.E.O. is a current employee, co-founder, and shareholder of UbiQ Bio BV. A.A. is a current employee of UbiQ Bio BV. C. S., B.B., and G.T.C. are current employees of Boston Biochem, Inc. M.B. declares no competing interests.

