## [Peer Review File(PDF 88 kb) · Nature Communications]

Reviewers' comments:

Reviewer #1 (Remarks to the Author):

In this manuscript, Hewings et. al. develop a mass spectrometry based approach to investigate cysteine's modified by ubiquitin activity-based probes. Using this approach, they identify ZUFSP as an unannotated deubiquitinase with selectivity for K63 linked chains when tetraubiquitin is utilized as the substrate. This well written and interesting report provides important new information about a new deubiquitinase while also developing a chemoproteomic approach to study DUB activity.

Concerns:

In Figure 6e and f the authors investigate the ability of ZUFSP to cleave either Ub4-Rh110 or Ub4-peptide-TAMRA. With Ub4-peptide-TAMRA the authors conclude that ZUFSP cleaves the middle of Ub chains with little production of Ub3, however, with Ub4-Rh110 Ub3 appears to be a major species. This discrepancy needs to be addressed.

Minor points:

I was unable to identify how long the DUB reactions in Figure 6a-c are.

The authors call ZUFSP a K63 specific DUB. While under the conditions tested the authors did see a clear preference for K63 linked chains it is possible that ZUFSP can cleave longer chains as dramatic increases in DUB activity have been observed for longer chains (e.g. ATXN3). Additionally, the authors have not tested all possible chain types (e.g. mixed chains, etc). The authors should consider using the term highly selective over specific.

Reviewer #2 (Remarks to the Author):

Ubiquitination and deubiquitination of proteins are crucial modifications for regulating cellular pathways and homeostasis. New targets and outcomes of ubiquitin modification are being discovered however the enzymes that regulate this modification are not always known. The manuscript by Hewings and coworkers describes a new method using activity based probes to

identify deubiquitinating enzyme:substrate pairs. The method appears to be as sensitive as previous methods but does not identify as many off-target proteins such as E2 and RING E3s as other methods. However, the authors report that for 5 deubiquitinating enzymes and some highly abundant proteins, they observed non-catalytic cysteine labeling. This observation shows the caution with which data from these types of studies should be interpreted. In an application of the method, the authors identify and characterize the activity of ZUFSP, a deubiquitinating enzyme which was previously suggested to be inactive.

Overall, the authors' work is of high quality and presents an exciting new strategy to expand the depth of the field. My main issue is with the conclusions regarding ZUFSP activity and the effects of the assay type on activity.

Major and minor concerns are listed below.

Major concerns

1.) The Ub-Rn110 and Ub-TAMRA results in Figure 6 appear to conflict between Figures 6e and 6f. The authors state in the last paragraph that the enzyme cleaves within the chain and not the proximal position which is supported by the prevalence of Ub2 in Figure 6f. However, in 6e, Ub3 appears first indicating that the distal-2Ub or 3Ub-proximal bond is cleaved and suggesting a change in specificity. This also could be a data presentation issue as the time points do not match between the figures. However, even at the first matching point, 30 minutes, there are some distinctions between the gels. I would like to know if this is a technical/assay issue where the probe on the proximal ubiquitin is affecting the activity enzyme or if this is two assays presented slightly differently which prevents correlation of the results.

2.) Based on the finding that the enzyme cleaves in the middle of the ubiquitin chain, there may be an issue in calculating the rate constant and K_m in Figure 6d. For the Rn110 to fluoresce the final isopeptide bond must be cleaved, which based on the data in 6f and Ub-AMC, is not the preferred site. In the assay in 6d, the final bond is cleaved in the presence of 3-fold excess monoubiquitin, one of the reaction products, which could mean that the enzyme is not in the steady-state. Since monoubiquitin is likely a competitive inhibitor then the K_m is likely higher and the k_{cat} is likely lower than reality. Another method of measuring the kinetics would be better or please clarify the experiment set up to indicate how the authors know the Rn110 fluorescence is occurring before build up of the product. Simply taking the initial rates is not enough since the assay detects only one of the possible cleavage events.

This issue also relates to the enzyme specificity issue described in 1 above.

Minor concerns

- 1.) Pg. 8, line starting "By contrast...probe labeling sites". The sentence is tough to read.
- 2.) k_{cat}/K_m values might be a better comparison for USP5 and ZUFSP if you are simply remarking on substrate quality.

Reviewer #3 (Remarks to the Author):

In the manuscript by Ingrid Wertz and colleagues, a novel approach to capture active cellular DUBs is described based on the use of active site probes especially designed to capture and release covalently bound enzyme species. This experimental approach is an extension of previously described chemical biology assays that make use of a mechanism-based covalent capture of cellular deubiquitylating enzymes (DUBs) from crude extracts. Since it has been described more than a decade ago, this approach has contributed to our understanding of the biological role of members of the DUB enzyme family, but has also permitted the evaluation of small molecule inhibitors with selectivity towards DUBs.

The combination of using ubiquitin active site probes with mass spectrometry as a means of discovering 'ub-probe' reactive DUBs has been described previously. However, one major limitation has been the direct assignment of DUBs that are directly bound to the probe or are co-immunoprecipitated via association with other proteins within a complex. As mass spectrometry has now become very sensitive and can detect ~1000 proteins pulled down in such experiments, it is now becoming more pressing to make more headway regarding this question, also for the discovery of novel enzymes with putative ubiquitin binding and hydrolase activity. The authors have used their new approach to characterise a novel DUB, ZUFSP, for which they demonstrate direct modification of the catalytic cysteine residue.

In view of the current interest in drug discovery within the ubiquitin system by academia and pharma, the present work definitively merits publication in this journal. However, there are a number of issues that should be addressed first before this manuscript can be considered for publication.

Page 10: Metalloprotease DUBs are often seen as DUBs pulled down with Cys-reactive probes. The authors state that this is mainly due to non-specific binding to beads. However, it could also be that metalloprotease DUBs (and possibly other proteins with ubiquitin binding domains in general) may specifically recognise the ubiquitin scaffold of the probe without reacting to the C-terminal electrophile moiety. This category of proteins (e.g. metalloprotease) DUBs should be mentioned and potentially listed separately for clarity.

Page 10/11: The authors show in their work that different cysteine residues labelled in DUBs such as USP7, USP10, USP14, USP22 and UCHL1. This is quite interesting, and the authors discuss this as a possible non-specific side effect of using active site directed probes. However, it is possible, for example shown in the case of OTUB1, that this reflects more than one ubiquitin bound to the DUB, in fact as many DUBs have multiple ubiquitin binding site. It is possible that

if another cysteine is located in proximity to the C-terminal electrophile of the probe bound to a second ubiquitin binding site, that additional reactions can occur. In the case of UCHL1, this seems to happen much less frequently as compared to binding to C90, the real catalytic site. This is reflected by the number of peptides found for C90 adducts as compared to other cysteine adducts (fig 4a).

Mass spectrometry data needs deposition in a publicly available resource, such as PRIDE or Proteome Commons. Many MS/MS spectra showing cysteine modifications are collected during this study, and there is little information about the quality and accuracy of the MS/MS assignments. In particular, claims of the Ub-probe labelling non-catalytic cysteine residues need to be based on solid MS/MS assignments, and this data should be made available for inspection, either by providing MS/MS spectra as supplementary information or as a public repository. This will make claims such as labelling "buried cysteines C47 and C220 by the probe more believable as they seem to be observed much less frequently as the modification of the catalytic cysteine 90 as the authors show in the case of UCHL1 (figure 4).

Responses to Reviewers:

We would like to thank all the reviewers for their helpful comments. Below is a point-by-point response to the issues raised.

Reviewers' comments:

Reviewer #1 (Remarks to the Author):

Concerns:

In Figure 6e and f the authors investigate the ability of ZUFSP to cleave either Ub₄-Rh110 or Ub₄-peptide-TAMRA. With Ub₄-peptide-TAMRA the authors conclude that ZUFSP cleaves the middle of Ub chains with little production of Ub₃, however, with Ub₄-Rh110 Ub₃ appears to be a major species. This discrepancy needs to be addressed.

The silver stain (Figure 6e) detects all ubiquitin species. By contrast, in Figure 6f, only proteins containing the proximal fluorophore are detected. Therefore, using the Ub₄-peptide-TAMRA substrate it is not possible to detect unconjugated Ub₃, but we would anticipate (based on the results in Figure 6e) that unconjugated Ub₃ is a significant product in this reaction. The lack of Ub₃-peptide-TAMRA suggests that ZUFSP does not cleave at the distal end of a chain, and instead requires at least two distal Ub monomers for activity.

We regret that this distinction was not made clear in the manuscript. To clarify this we added the following sentence: "ZUFSP can clearly process Ub₄-containing species to Ub₃ (Figure 6a), but Ub₃ formed by cleavage between the two proximal Ub moieties of Ub₄-peptide-TAMRA will not be detected by this method, since it lacks a fluorophore."

Minor points:

I was unable to identify how long the DUB reactions in Figure 6a-c are.

The legend for Figure 6 now includes the reaction times.

The authors call ZUFSP a K63 specific DUB. While under the conditions tested the authors did see a clear preference for K63 linked chains it is possible that ZUFSP can cleave longer chains as dramatic increases in DUB activity have been observed for longer chains (e.g. ATXN3). Additionally, the authors have not tested all possible chain types (e.g. mixed chains, etc). The authors should consider using the term highly selective over specific.

'Specific has been replaced with 'selective'.

Reviewer #2 (Remarks to the Author):

Major concerns

1.) The Ub-Rn110 and Ub-TAMRA results in Figure 6 appear to conflict between Figures 6e and 6f. The authors state in the last paragraph that the enzyme cleaves within the chain and not the proximal position which is supported by the prevalence of Ub₂ in Figure 6f. However, in 6e, Ub₃ appears first indicating that the distal-2Ub or 3Ub-proximal bond is cleaved and suggesting a change in specificity. This also could be a data presentation issue as the time points do not match between the figures. However, even at the first matching point, 30 minutes, there are some distinctions between the gels. I would like to know if this is a technical/assay issue where the probe on the proximal ubiquitin is affecting the activity enzyme or if this is two assays presented slightly differently which prevents correlation of the results.

Please see response to Reviewer 1.

2.) Based on the finding that the enzyme cleaves in the middle of the ubiquitin chain, there are may be issue in calculating the rate constant and Km in Figure 6d. For the Rn110 to fluoresce the final isopeptide bond must be cleaved, which based on the data in 6f and Ub-AMC, is not the preferred site. In the assay in 6d, the final bond is cleaved in the presence of 3-fold excess monoubiquitin, one of the reaction products, which could mean that the enzyme is not in the steady-state. Since monoubiquitin is likely a competitive inhibitor then the Km is likely higher and the kcat is likely lower than reality. Another method of measuring the kinetics would be better or please clarify the experiment set up to indicate how the authors know the Rn110 fluorescence is occurring before build up of the product. Simply taking the initial rates is not enough since the assay detects only one of the possible cleavage events.

This issue also relates to the enzyme specificity issue described in 1 above.

We agree with the reviewer that there are some caveats to inferring kinetic data for ZUFSP with the Ub₄-Rh110 substrate, which reflect the inevitable challenges of determining kinetic parameters where an enzyme can process a substrate in multiple ways. An ideal substrate for studying ZUFSP kinetics would have only a single site that can be processed by the enzyme.

The reviewer suggests that the data from Ub-AMC and Ub₄-peptide-TAMRA experiments indicate that the Ub-Rh110 bond is not the preferred cleavage site, but we do not think the data supports this conclusion. Firstly, Ub-AMC is likely to be a poor substrate because it contains only a single Ub moiety, and ZUFSP recognizes and preferentially processes long chains. Secondly, the Ub₄-peptide-TAMRA substrate contains an isopeptide bond between the proximal Ub, while Ub₄-Rh110 contains a peptide bond, which may be processed differently.

If there were substantial product inhibition or other more complex behavior, we would expect to see biphasic kinetics in the raw fluorescence data, which we do not observe. Supplementary Figure 18 does indeed suggest that mono-Ub is a competitive inhibitor, but only at very high concentrations (>50 μM), which cannot build up during the initial stage of the assay. (The starting Ub₄-

Rh110 concentration in the assay was in the range 25 μ M to 625 nM).

To clarify these points, we have added the following passage to the figure legend (Supporting Figure 17): “Note that this method only detects the hydrolysis of the proximal Ub-fluorophore bond, not internal sites. Since biphasic kinetics are not observed, it is likely that at early time points the measured rates reflect hydrolysis at the proximal end of the tetra-Ub chain, not hydrolysis of smaller Ub-Rh110 conjugates (or that these substrates are processed at similar rates).” Furthermore, we emphasize in the text that we investigated Ub₄-Rh110 as a potential substrate in order to facilitate further biochemical studies (such as the competition experiments show in Supplementary Figure 18), rather than to obtain fundamental insights into ZUFSP behavior.

Minor concerns

1.) Pg. 8, line starting "By contrast...probe labeling sites". The sentence is tough to read.

This now reads: “By contrast, the acid-labile dialkoxydiphenylsilane (DADPS) linker²⁵ is suitable for on-bead digestion and identification of probe labeling sites, since labeled peptides could be detected following enrichment and DADPS cleavage (Figure 2c, d).”

2.) kcat/Km values might be a better comparison for USP5 and ZUFSP if you are simply remarking on substrate quality.

We agree that comparing Ub₄-Rh110 processing by ZUFSP and USP5 suggests that Ub₄-Rh110 is a poorer substrate for ZUFSP, and does not necessarily tell us anything about intrinsic catalytic activity of the two enzymes. We now state only: “ZUFSP processed K63-linked Ub₄-rhodamine 110 (Ub₄-Rh110), although it showed lower activity towards this substrate than USP5 (Supplementary Figure 17), which is highly active towards K63-linked chains.”

Reviewer #3 (Remarks to the Author):

Page 10: Metalloprotease DUBs are often seen as DUBs pulled down with Cys-reactive probes. The authors state that this is mainly due to non-specific binding to beads. However, it could also be that metalloprotease DUBs (and possibly other proteins with ubiquitin binding domains in general) may specifically recognise the ubiquitin scaffold of the probe without reacting to the C-terminal electrophile moiety. This category of proteins (e.g. metalloprotease) DUBs should be mentioned and potentially listed separately for clarity.

Non-covalent interactions between the probe and proteins with Ub-binding domains could indeed lead to enrichment. However, the extremely stringent washing conditions employed in our method (2% SDS, 8 M urea, 1 M NaCl) make this unlikely. Furthermore, metalloprotease DUBs are not significantly enriched by Ub-VPS over two negative controls: omission of copper during

biotinylation, or the use of the SUMO2-VPS probe. If the interaction between the metalloprotease DUB and the probe were non-covalent, we would expect these DUBs to be enriched by Ub-VPS relative to SUMO2-VPS. Therefore these DUBs are probably detected due to non-specific interactions. We have amended the discussion to clarify this. (With the traditional HA-Ub-VS probe and IP enrichment, the enrichment of metalloproteases could be due to non-covalent interactions with the probe or with the beads.)

Page 10/11: The authors show in their work that different cysteine residues labelled in DUBs such as USP7, USP10, USP14, USP22 and UCHL1. This is quite interesting, and the authors discuss this as a possible non-specific side effect of using active site directed probes. However, it is possible, for example shown in the case of OTUB1, that this reflects more than one ubiquitin bound to the DUB, in fact as many DUBs have multiple ubiquitin binding site. It is possible that if another cysteine is located in proximity to the C-terminal electrophile of the probe bound to a second ubiquitin binding site, that additional reactions can occur. In the case of UCHL1, this seems to happen much less frequently as compared to binding to C90, the real catalytic site. This is reflected by the number of peptides found for C90 adducts as compared to other cysteine adducts (fig 4a).

This is an excellent point. Most labeling sites are found within the catalytic domain (which could itself interact with multiple Ub units), but some are in other parts of the protein. This is the case for C219 of USP5, which is found in a ubiquitin-interacting zinc finger and is positioned close to the binding site of the C-terminus of Ub. We have now highlighted this in the text.

Mass spectrometry data needs deposition in a publicly available resource, such as PRIDE or Proteome Commons. Many MS/MS spectra showing cysteine modifications are collected during this study, and there is little information about the quality and accuracy of the MS/MS assignments. In particular, claims of the Ub-probe labelling non-catalytic cysteine residues need to be based on solid MS/MS assignments, and this data should be made available for inspection, either by providing MS/MS spectra as supplementary information or as a public repository. This will make claims such as labelling "buried cysteines C47 and C220 by the probe more believable as they seem to be observed much less frequently as the modification of the catalytic cysteine 90 as the authors show in the case of UCHL1 (figure 4).

Mass spectrometry raw datasets and peptide-spectrum matches have been deposited in MassIVE (a member of the ProteomeXchange Consortium). They can be accessed using username 'MSV000081972_reviewer' and password 'gne2018' by this url: <https://massive.ucsd.edu/ProteoSAFe/dataset.jsp?task=72deff96dd7d4c6f86dd1e5f7e5ee265>. Upon publication, we would make the datasets publically available.

Additionally, we have included a number of MS/MS spectra in the manuscript as examples, to support the assignment of UCHL1 and ZUFSP labeling sites.

Reviewers' Comments:

Reviewer #2 (Remarks to the Author):

The authors have addressed my concerns.

Reviewer #3 (Remarks to the Author):

The authors have revised their manuscript based on the criticism raised by the reviewers. In particular, the issues about Figure 6 and the mass spectrometry data supporting modification of non-catalytic cysteines have been addressed. There are a number of non-DUB and non-E3 proteins found to contain cysteine residues modified by this reactive site centric ubiquitin probe (Table 1). If proven to be correct, it will be interesting to see whether these proteins have intrinsic ubiquitin binding sites adjacent to reactive cysteine residues.

I am happy with the revised manuscript and would now definitively recommend it for publication.